# Apical annuli are specialised sites of post-invasion secretion of dense granules in *Toxoplasma*

**Sara Chelaghma†, Huiling Ke†, Konstantin Barylyuk, Thomas Krueger, Ludek Koreny\*, Ross F Waller\***

Department of Biochemistry, University of Cambridge, Cambridge, United Kingdom

**Abstract** Apicomplexans are ubiquitous intracellular parasites of animals. These parasites use a programmed sequence of secretory events to find, invade, and then re-engineer their host cells to enable parasite growth and proliferation. The secretory organelles micronemes and rhoptries mediate the first steps of invasion. Both secrete their contents through the apical complex which provides an apical opening in the parasite's elaborate inner membrane complex (IMC) – an extensive subpellicular system of flattened membrane cisternae and proteinaceous meshwork that otherwise limits access of the cytoplasm to the plasma membrane for material exchange with the cell exterior. After invasion, a second secretion programme drives host cell remodelling and occurs from dense granules. The site(s) of dense granule exocytosis, however, has been unknown. In *Toxoplasma gondii*, small subapical annular structures that are embedded in the IMC have been observed, but the role or significance of these apical annuli to plasma membrane function has also been unknown. Here, we determined that integral membrane proteins of the plasma membrane occur specifically at these apical annular sites, that these proteins include SNARE proteins, and that the apical annuli are sites of vesicle fusion and exocytosis. Specifically, we show that dense granules require these structures for the secretion of their cargo proteins. When secretion is perturbed at the apical annuli, parasite growth is strongly impaired. The apical annuli, therefore, represent a second type of IMC-embedded structure to the apical complex that is specialised for protein secretion, and reveal that in *Toxoplasma* there is a physical separation of the processes of pre- and post-invasion secretion that mediate host-parasite interactions.

**\*For correspondence:**
lk360@cam.ac.uk (LK);
rfw26@cam.ac.uk (RFW)

†These authors contributed equally to this work

**Competing interest:** The authors declare that no competing interests exist.

## Editor's evaluation

This important study identifies a novel mechanism for protein secretion in the obligate human protozoan parasite *Toxoplasma gondii*. The authors utilize a range of advanced imaging, proteomic and genetic approaches to convincing demonstrate the mechanism by which dense granule proteins are delivered to the parasite plasma membrane. This study will be of broad interest to cell biologists and parasitologists working on membrane trafficking and virulence mechanisms.

## Introduction

Apicomplexa is a phylum of ubiquitous eukaryotic parasites of animals of broad medical, veterinary, and ecological importance. In humans they are responsible for diseases such as malaria, cryptosporidiosis, and toxoplasmosis causing widespread mortality and morbidity which disproportionately affects developing world regions (*Havelaar et al., 2015*; *Montoya and Liesenfeld, 2004*; *Striepen, 2013*; *WHO, 2021*). Commercial live-stock industries and subsistence farming also suffer heavy losses from diseases caused by apicomplexans, including coccidiosis in poultry, babesiosis, and theileriosis

in cattle, and fetal death in sheep and goats from toxoplasmosis (*MacGregor et al., 2021*). Apicomplexans belong to a larger group of related unicellular eukaryotes including ecologically important dinoflagellates and ciliates that together form the supergroup Alveolata. Common to these organisms is a complex cell pellicle derived from a tessellation of flattened membrane alveolar vesicles appressed to the inner face of the cell's plasma membrane and supported by a complex proteinaceous membrane skeleton (*Anderson-White et al., 2012*; *Gould et al., 2008*). This cell pellicle provides broad structural and functional roles to these cells. In apicomplexans the pellicle defines their distinctive cell shape and provides a platform for both gliding motility machinery and signalling networks, all of which are critical to the parasites' ability to navigate their animal host tissues and invade selected cells.

A common challenge posed to alveolate cells by their complex pellicle is that it limits opportunities for material exchange across the plasma membrane. The alveolar vesicles and subpellicular proteinaceous networks, collectively called the inner membrane complex (IMC) in apicomplexans, typically line most of the inner cell surface. Dedicated structures within the IMC are, therefore, required to provide access to the plasma membrane for processes such as exocytosis and endocytosis. In apicomplexans an apical complex is one such structure that provides an apical opening in the IMC to allow exocytosis from two invasion-related organelles: micronemes and rhoptries (*Koreny et al., 2021*; *Dos Santos Pacheco et al., 2020*). Secretion of microneme proteins promotes parasite egress from its host cell, gliding motility, and then apical contact with its next host cell to invade. At this point, secretion from rhoptries facilitates host cell entry and some countering of the host cell's immune response (*Hakimi et al., 2017*). The apical complex is composed of an apical polar ring of proteins, that excludes the IMC, and within which sits a hollow tubulin-based conoid. The micronemes and rhoptries extend through the conoid to reach the free plasma membrane at this apical site. The requirement for microneme and rhoptry secretion for invasion makes this an essential structure to these parasites. A second dedicated type of structure in the IMC allows material transfer in the opposite direction. Micropores are small invaginations of the plasma membrane that are supported by a collar of proteins positioned at the intersections of IMC alveolar vesicles (*Nichols et al., 1994*). While these structures were first observed in early electron microscopy on these cells, their role in endocytosis has been only recently established (*Koreny et al., 2023*; *Wan et al., 2023*). Both the apical complex and micropores are found throughout apicomplexans, as well as in related dinoflagellates, perkinsids, and chrompodelids, which suggests that they have been fundamental features of the evolution of the complex pellicle of Alveolata.

A third, enigmatic type of annular structure has been observed within the IMC of the apicomplexan *Toxoplasma gondii*. This structure occurs as a series of ~5–8 small annuli in a ring around the periphery of the cell but towards its apical end (*Engelberg et al., 2020*; *Hu et al., 2006*). Originally named peripheral annuli, but now more commonly called apical annuli, these structures were previously known only from protein location patterns as rings but were otherwise cryptic in ultrastructural studies. Electron microscopy on shadow-cast detergent extractions of *T. gondii* pellicles, however, do reveal small rings, intercalated between the subpellicular network of microtubules at the expected position for the apical annuli (*Diaz-Martin et al., 2022*). The proteins initially characterised at the apical annuli lacked membrane-binding domains and appeared to interact with proteins of the IMC subpellicular network including 'suture' proteins that occur where alveolar vesicles are sealed together (*Engelberg et al., 2020*; *Hu et al., 2006*; *Suvorova et al., 2015*). This implied that the apical annuli were features of the IMC occurring at alveolar junctions, but it was not known if these structures interacted with or were relevant to the plasma membrane including the cell surface. Mutations of proteins specific to the apical annuli lacked clear phenotypes that could inform on the significance or function of these structures (*Engelberg et al., 2020*). Therefore, it has been a mystery what the apical annuli might do in these cells.

A second conundrum of the apicomplexan pellicle has been where does the third major class of exocytic vesicles, the so-called dense granules, exocytose from the cell? Dense granule proteins are secreted after the parasite has invaded its new host (*Carruthers and Sibley, 1997*). They deliver a complex mixture of proteins that establish the parasitophorous vacuole in which the parasite resides in its host, and also target a wide range of host cell organelles to re-engineer this environment in which it grows (*Griffith et al., 2022*; *Lebrun et al., 2014*). These functions include interfering with host signalling networks and transcriptional programmes that would otherwise seek to attack the intruder

(*Hakimi et al., 2017*). The identification of dense granule organelles throughout Apicomplexa is more challenging because they lack the conspicuous morphologies that characterise micronemes and rhoptries, and they tend to have fast-evolving and unique protein repertoires specific to each parasite-host interaction (*Barylyuk et al., 2020*; *Guérin et al., 2023*). Nevertheless, it is presumed that most apicomplexans rely on equivalent post-invasion secretion. Dense granules also differ from micronemes and rhoptries, which are permanently concentrated around the cell apex, in that they are dispersed throughout the cell. They are highly motile on cytoskeletal networks, and they have been seen to accumulate towards the apex during invasion events (*Heaslip et al., 2016*; *Venugopal et al., 2020*). While some evidence of their secretion from the apical region of the cell has been proffered, the precise sites or mechanism of dense granule exocytosis has remained unknown (*Dubremetz et al., 1993*).

In our study, we have asked if any specific plasma membrane proteins occur at the apical annuli sites in *T. gondii* and, if so, could these inform on apical annuli function including the processes of post-invasion dense granule protein secretion?

## Results

### The apical annuli structures span the parasite plasma membrane

The question of whether the apical annuli structures of *T. gondii* extend to include the plasma membrane was fortuitously answered by our independent investigations of *T. gondii* membrane proteins at the cell surface. During the verification of protein location assignments from our hyperLOPIT spatial proteomic studies, we C-terminally reporter-tagged an uncharacterised protein that we call *Tg*LMBD3 (TGME49_222200) that was assigned as a transmembrane protein of the plasma membrane by hyper-LOPIT (*Barylyuk et al., 2020*). By immunofluorescence assay (IFA) the endogenously reporter-tagged *Tg*LMBD3 showed a distinctive surface location pattern of ~5–8 puncta arranged as a subapical ring at the cell surface (*Figure 1A*). Centrin2 is a known marker of the apical annuli, although it additionally occurs at discrete locations at the conoid, centrosome, and basal complex (*Hu et al., 2006*; *Leung et al., 2019*). *Tg*LMBD3 was exclusively located to the Centrin2 apical annuli positions (*Figure 1A*). The predicted membrane topology of this new protein includes nine transmembrane domains with a cytosolic C-terminal extension (*Figure 1B*). To verify the plasma membrane location of this protein, we performed trypsin proteolytic shaving of live parasites. *Tg*LMBD3, which migrates as a double band on SDS-polyacrylamide gel electrophoresis, showed rapid trypsin sensitivity (*Figure 1C*). Similarly, the surface protein SAG1 was progressively diminished as internal SAG1 pools recycled to the surface (*Gras et al., 2019*; *Koreny et al., 2023*), whereas internal marker proteins (GAP40, PRF, TOM40) were all trypsin-protected by the plasma membrane. These data are consistent with exposure of inter-transmembrane loops of *Tg*LMBD3 at the cell surface and, therefore, that the apical annuli are structures that extend to the surface environment of the cell.

Can this new plasma membrane protein inform us of apical annuli function? *Tg*LMBD3 contains a conserved limb development membrane protein 1 domain (LMBD) that consists of 9 transmembrane helices in a 5+4 arrangement (*Figure 1B*) but whose functions are generally not well studied (*Redl and Habeler, 2022*). Furthermore, multiple paralogous LMBD proteins exist in eukaryotes, so we first asked if *Tg*LMBD3 belonged to an orthogroup with any functional characterisation. A global phylogeny of LMBD proteins shows that four LMBD orthogroups exist (*Figure 1—figure supplement 1A*). *Tg*LMBD3 belongs to orthogroup III and, hence, our nominated name for this protein. Although orthogroup III is widely present in eukaryotes, all animals (holozoa) have lost this paralogue, and none have been functionally characterised. Nevertheless, orthologues of *Tg*LMBD3 are retained in all apicomplexans, and throughout their close alveolate relatives (including dinoflagellates and ciliates) (*Figure 1—figure supplement 1B*). Thus, while a function for *Tg*LMBD3 cannot be readily identified from orthologues, its strong conservation throughout Apicomplexa implies that it performs an important function in *Toxoplasma*.

### Apical annuli occur at gaps in the IMC

Given that *Tg*LMBD3 implicates the plasma membrane in apical annuli function, we asked if the IMC is coordinated with the apical annuli sites to allow interaction here between the plasma membrane and the rest of the cell cytosol. We first sought higher resolution position information for *Tg*LMBD3

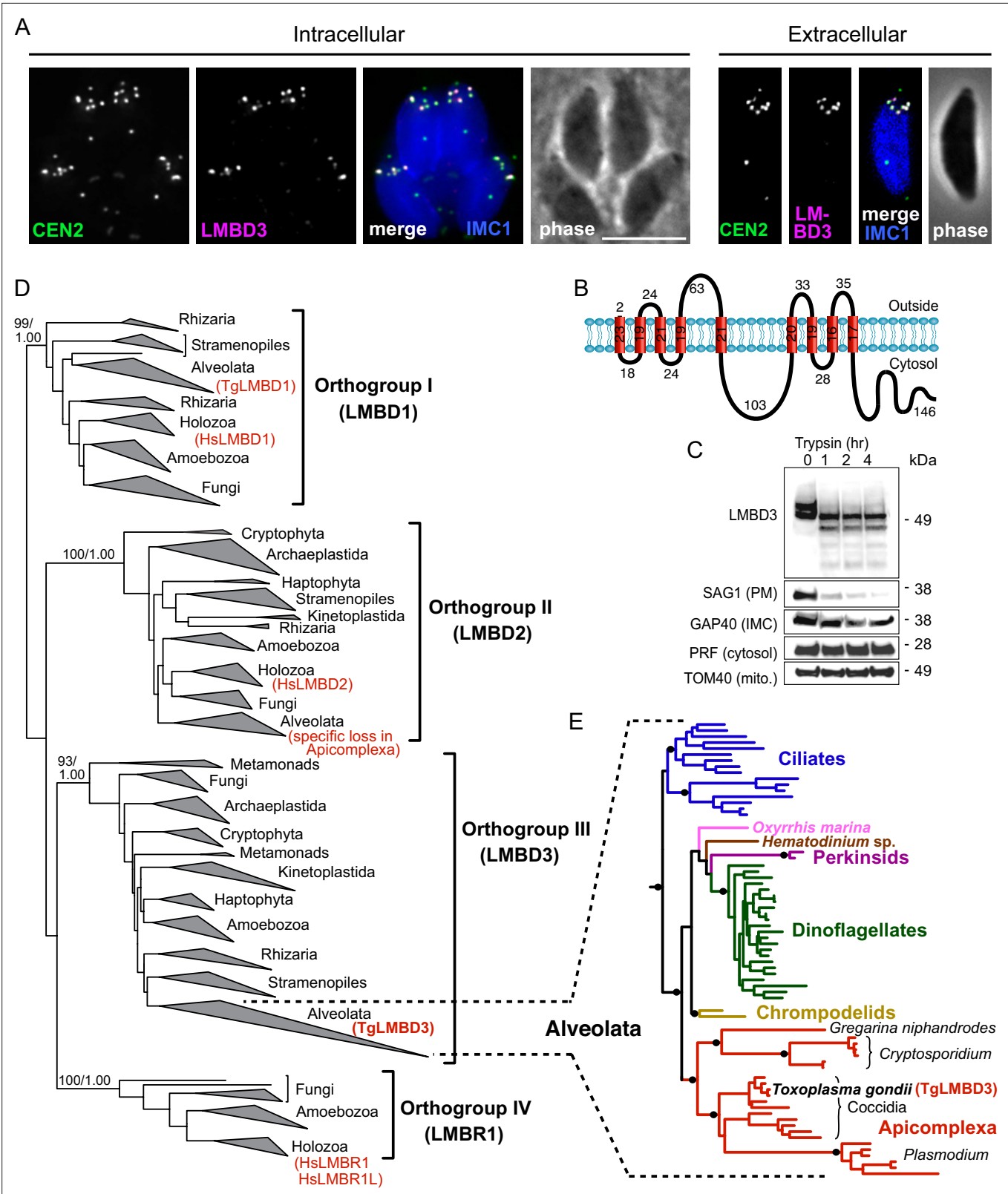

**Figure 1.** *Tg*LMBD3 is a conserved protein in the plasma membrane at apical annuli sites in *T. gondii*. (**A**) Wide-field immunofluorescence assay imaging of cells expressing *Tg*LMBD3-6xHA (magenta) and eGFP-Centrin2 (green) and immunostained inner membrane complex (IMC)1 (blue) in either the intracellular stage in hosts or extracellular tachyzoites. Scale bar=5 μm. (**B**) Membrane topology of *Tg*LMBD3 by DeepTMHMM (*Hallgren et al., 2022*) with numbers indicating amino acid domain lengths. (**C**) Trypsin-shaving sensitivity over 4 hr, visualised on western blots, of *Tg*LMBD3-6xHA and markers

*Figure 1 continued on next page*

*Figure 1 continued*

of the exterior leaflet of the plasma membrane (PM), IMC, cytosol (PRF, profilin), and mitochondrion (mito.). kDa, kilodalton. (**D**) Maximum likelihood phylogeny of limb development membrane protein 1 domain (LMBD) proteins resolving as four major orthogroup clades. Node support values are bootstraps followed by aLRT SH-like supports. (**E**) Expanded Alveolata clade from LMBD orthogroup III showing major groups. Black dots indicate aLRT SH-like support >0.95. See *Figure 1—figure supplement 1* for full phylogenies.

The online version of this article includes the following source data and figure supplement(s) for figure 1:

**Figure 1-source data 1** .Trypsin-shaving sensitivity over four hours (hr), visualised on Western blots of TgLMBD3-6xHA extracts using antibodies against HA, Tom40, Sag1, profilin (Prf) and Gap40.

**Figure supplement 1.** Maximum likelihood phylogenies of the LMBR1 domain-containing proteins from *Figure 1*.

relative to the small rings that Centrin2 delineates. Three-dimensional structured illumination microscopy (3D-SIM) resolved *Tg*LMBD3 as a small punctum within the Centrin2 rings although slightly peripheral to it, consistent with *Tg*LMBD3 being in the plasma membrane and Centrin2 associated with the IMC beneath (*Figure 2A*). We then asked how *Tg*LMBD3 is positioned with respect to the IMC cisternae boundaries by staining the IMC suture protein ISC3 (*Chen et al., 2017*). The *Tg*LMBD3-labelled annuli were always positioned at the cisternae boundaries, specifically where the apical cap boundary intersects with the longitudinal suture boundaries (*Figure 2B*). All such intersections observed contained a *Tg*LMBD3 punctum and, hence, the variable number of apical annuli from cell to cell is apparently defined by the number of longitudinal IMC boundaries. The IMC is supported on its plasma membrane-facing surface by GAP45, and on its cytosol-facing surface by IMC1 (*Anderson-White et al., 2011*). While both proteins occur throughout the peripheral IMC regions of the cell, both showed an interruption where the *Tg*LMBD3 puncta occur (*Figure 2C and D*). Centrin2, and other markers of the apical annuli, are recruited to the IMC during early stages of daughter cell formation (*Engelberg et al., 2020*). In apicomplexans, these early cell stages lack the plasma membrane which is only recruited when the new cells emerge from within the mother cell – the process of endodyogeny in *Toxoplasma*. Given that *Tg*LMBD3 is a plasma membrane protein, we asked when is this protein recruited to the apical annuli? Cells captured throughout the process of endodyogeny showed that *Tg*LMBD3 only appears at apical puncta as the cell apex emerges from the mother cell at the time of plasma membrane recruitment (*Figure 3*). Relicts of the mother cell's apical annuli *Tg*LMBD3 persist at this stage (*Figure 3*, arrows), suggesting that little or no *Tg*LMBD3 is recycled from mother to daughter. Collectively, these data imply that the apical annuli provide coordinated gaps in the IMC barrier that forms at the earliest point of IMC development and that they maintain access of the cytosol to these specialised locations in the plasma membrane.

## Apical annuli recruit SNARE proteins

Because *Tg*LMBD3 did not provide an obvious clue to why the plasma membrane should be accessible at the apical annuli, we asked if there are other plasma membrane-associated proteins at these sites. We employed a proximity-dependant biotinylation approach (BioID) using *Tg*LMBD3 as a promiscuous biotin ligase (BirA*)-conjugated bait. This approach recovered known apical annuli proteins as the most significantly BioID-enriched proteins (e.g. AAP2–5, *Supplementary file 1*). However, in pursuit of plasma membrane-associated proteins, we looked for any detected biotinylated proteins with hyperLOPIT location assignments as integral plasma membrane proteins (*Supplementary file 1*, *Barylyuk et al., 2020*). Amongst these proteins were three SNARE (soluble N-ethylmaleimide-sensitive factor attachment protein receptor) proteins, each containing either a Qa SNARE domain (TGME49_209820), Qb SNARE domain (TGME49_306640), or Qc SNARE domain (TGME49_253360). These three domain-containing proteins typically function at a target membrane of vesicle fusion by forming heterotetrameric complexes with a vesicle-bound R SNARE which drives the fusion of the vesicle (*Figure 4A*, *Jahn and Scheller, 2006*). Molecular phylogenies place these three proteins with canonical SNARE orthologues: the Qa SNARE with 'SyntaxinPM' orthologues, the Qb SNARE with 'NPSN' orthologues, and the Qc SNARE with 'Syp7' orthologues (*Dacks and Doolittle, 2002*; *Klinger et al., 2022*; *Venkatesh et al., 2017*). We accordingly propose the names: *Tg*StxPM (TgME49_209820), *Tg*NPSN (TGME49_306640), *Tg*Syp7 (TGME49_253360) based on their orthologies. To test if the three Q SNAREs occur at the apical annuli, each protein was endogenously N-terminally 3xV5 reporter-tagged (C-terminal fusions are not tolerated by these tail-anchored membrane proteins) and their locations visualised by IFA. All three SNAREs showed distinct apical annuli locations (*Figure 4B*).

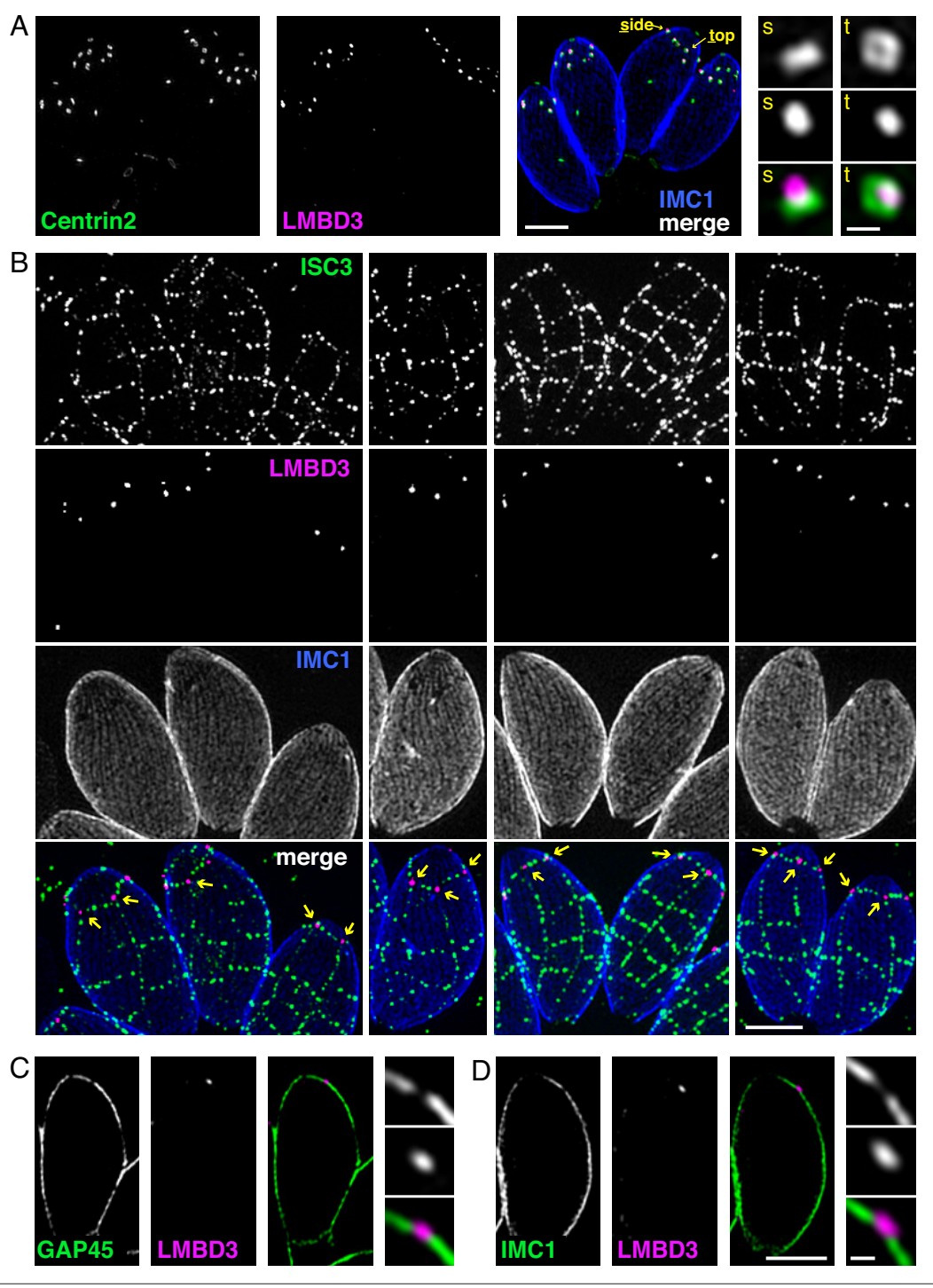

**Figure 2.** Apical annuli occur at gaps in the inner membrane complex (IMC). Three-dimensional structured illumination microscopy (3D-SIM) imaging of immunofluorescence assays of intracellular tachyzoites in host cells. (**A**) *Tg*LMBD3-6xHA (magenta) and eGFP-Centrin2 (green) expressing cells with side (**s**) and top (**t**) projections of apical annuli shown in zoom. IMC1 (blue). (**B**) Suture protein ISC3-3xV5 (green) co-expressed with *Tg*LMBD3-6xHA (magenta) showing apical annuli positioned at the apical cap suture where it intersects with longitudinal sutures (arrows). IMC1 (blue). (**C, D**) Optical sections showing GAP45 or IMC1 (green) with *Tg*LMBD3-6xHA (magenta) showing gaps in these IMC proteins where apical annuli occur. All scale bars=2 μm or 200 nm for zoomed panels.

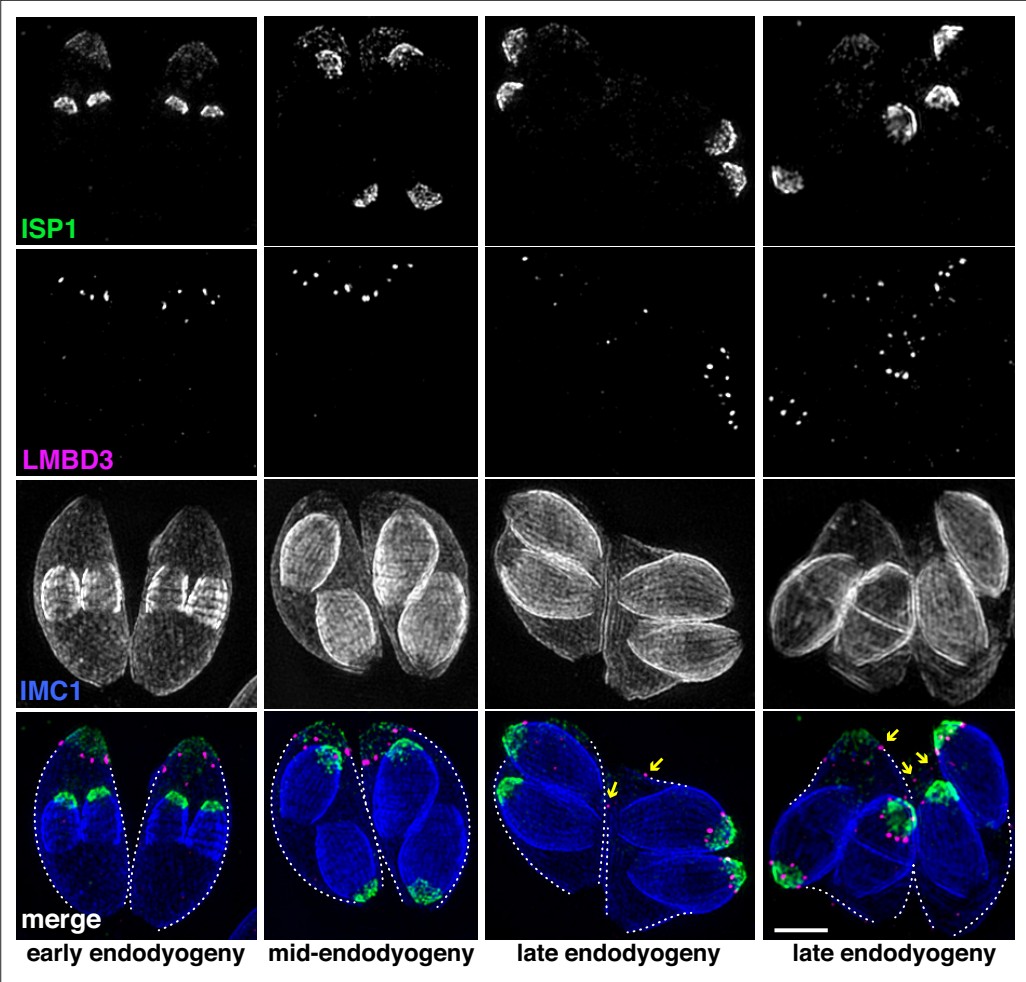

**Figure 3.** *Tg*LMBD3 is recruited to the apical annuli late in cytokinesis. Three-dimensional structured illumination microscopy (3D-SIM) imaging of immunofluorescence assays of intracellular tachyzoites at various stages of daughter cell formation and emergence from the mother cell. Maternal annuli are indicated with arrows. *Tg*LMBD3-6xHA (magenta) co-labelled with apical cap marker ISP1 (green) and inner membrane complex (IMC)1 (blue). Scale bar=2 μm.

*Tg*NPSN is exclusively located at apical annuli, whereas *Tg*StxPM and *Tg*Syp7 show some additional signal as a single structure in the central region of the cell (*Figure 4B*). Super resolution imaging of *Tg*NPSN shows a similar location to *Tg*LMBD3 relative to Centrin2: a small punctum centred on the Centrin2 ring but displaced towards the plasma membrane (*Figure 4C*). Together, these SNAREs implicate the annuli as sites for exocytic vesicle fusion at the plasma membrane.

## Apical annuli provide essential function for normal cell growth

Mutations of previously identified apical annuli proteins have shown either no phenotype or phenotypes that could not be discerned from possible additional functions of proteins, such as Centrin2 that occur at non-apical annuli cell sites also (*Engelberg et al., 2020*; *Lentini et al., 2019*; *Leung et al., 2019*). So, the importance of these cell structures to cell fitness was unknown. Using the four plasma membrane-associated apical annuli proteins we asked if these structures are required for typical cell growth. We made auxin-inducible degron (AID) knockdown cell lines (using the mini (m) AID [mAID] peptide) for the four proteins: *Tg*LMB3mAID-3xV5, mAID-3xV5-*Tg*StxPM, mAID-3xV5-*Tg*NPSN, and mAID-3xV5-*Tg*Syp7, and tested for growth phenotypes. All proteins were depleted to undetectable levels within 3–12 hr of auxin (3-indolacetic acid [IAA]) treatment (*Figure 5A*). Plaque assays assess tachyzoite competence for the entire lytic cycle, and the depletion of all four proteins showed strong phenotypes of reduced plaque development (*Figure 5B*). We then tested specifically for effects of

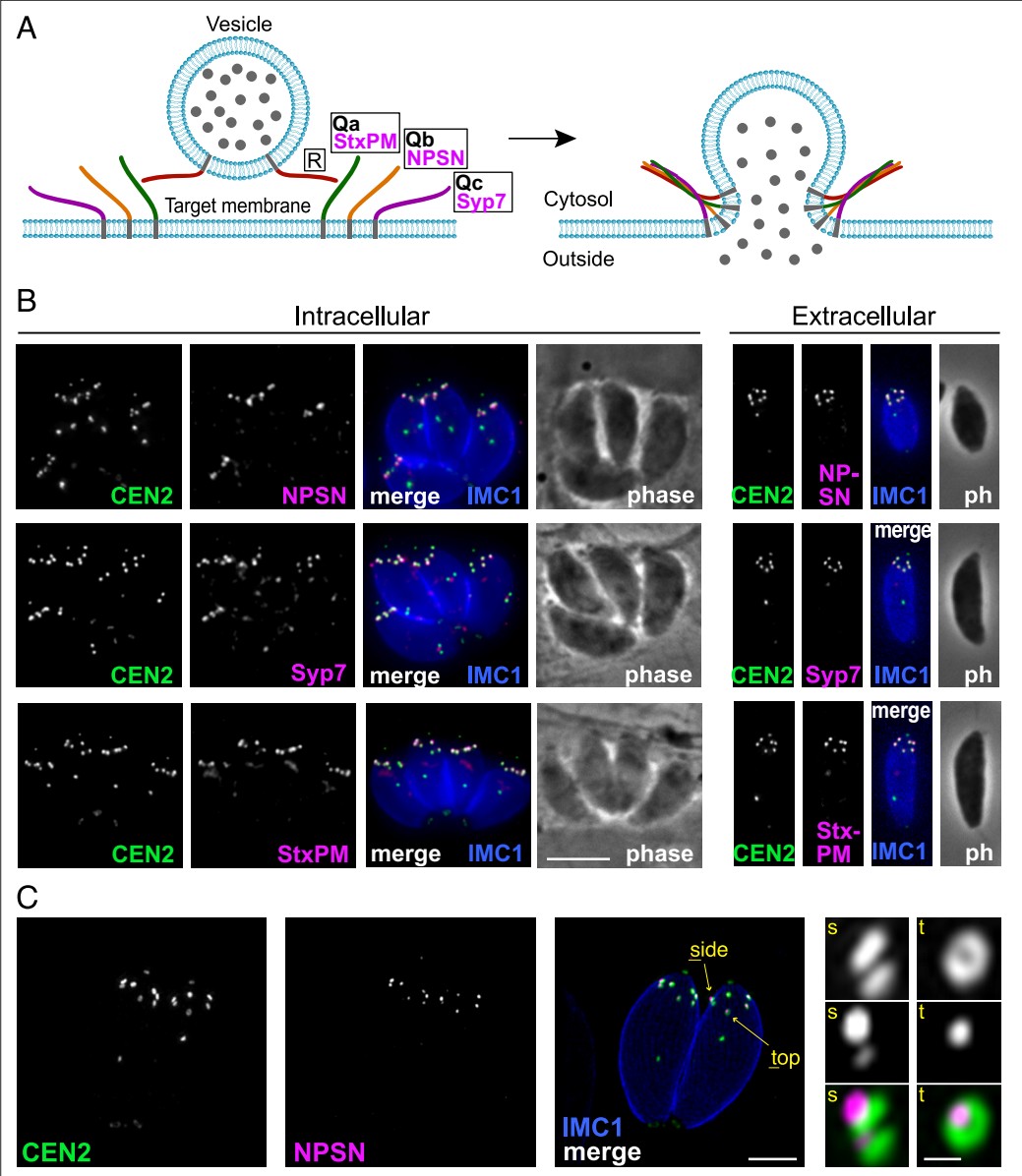

**Figure 4.** Three SNARE proteins likely form a complex at the inner side of the plasma membrane at the apical annuli. (**A**) Schematic of the SNARE complex which facilitates fusion of a secretory vesicle with target membrane (*Jahn and Scheller, 2006*). (**B**) Wide-field fluorescence microscopy localisation of *Tg*StxPM, *Tg*NPSN, and *Tg*Syp7 SNAREs expressed in *T. gondii* (as 3xV5 N-terminal fusions) and co-stained for eGFP-Centrin2. All panels are in the same magnification with scale bar=5 μm. (**C**) Three-dimensional structured illumination microscopy (3D-SIM) image of *Tg*NPSN with Centrin2. The zoomed panels show the apical annuli either in side (**s**) or top (**t**) projection as indicated. The scale bars for large and small (zoomed) panels are 2 μm and 200 nm, respectively.

each protein knockdown on parasite replication within the host cell. Depletion of all proteins resulted in delayed parasite replication at 24 hr post invasion (*Figure 5C*). All mutants showed an average lag of one to three division cycles behind the control, with the depletion of *Tg*StxPM showing the most severe retardation of parasite replication. These data show that the apical annuli are necessary for cell proliferation in the host cell environment.

## Apical annuli are required for dense granule exocytosis

Micronemes and rhoptries release their contents during the initial events of host cell invasion by fusing with the plasma membrane accessed through the conoid at the cell's apex (*Aquilini et al., 2021*;

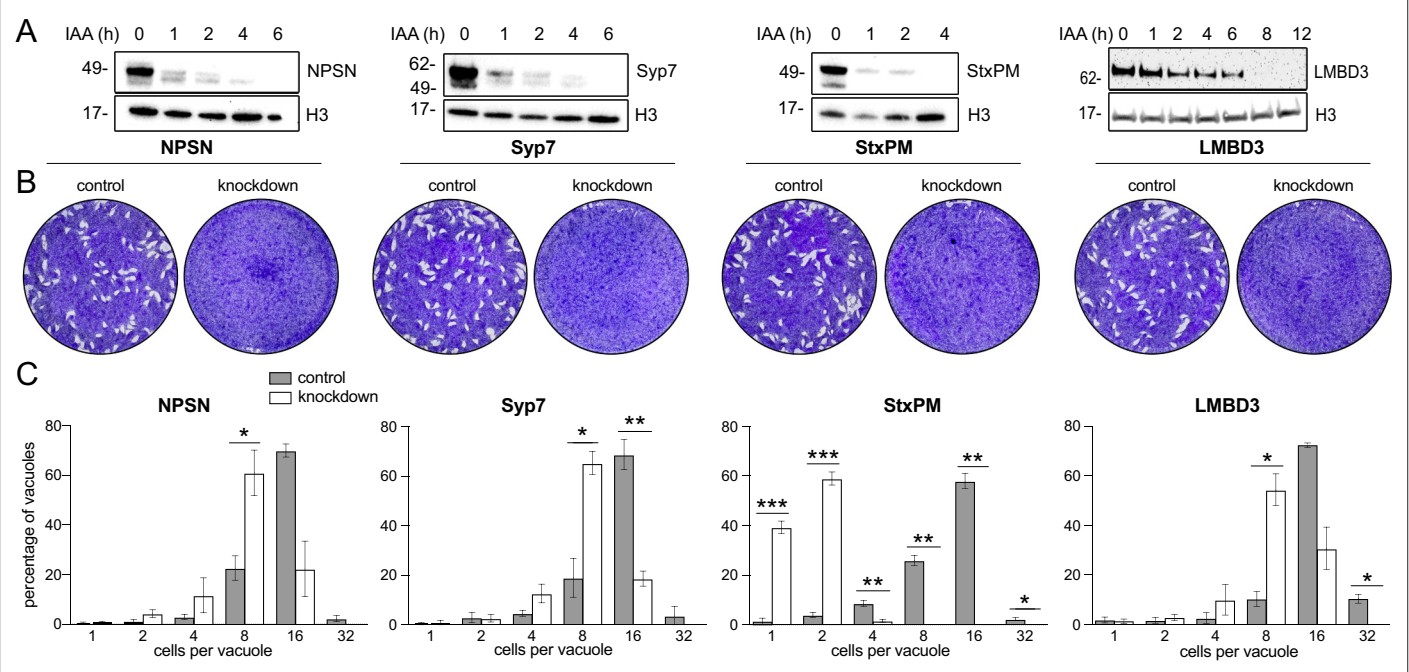

**Figure 5.** Depletion of apical annuli plasma membrane proteins impairs replication rates of *T. gondii*. (**A**). Depletion of each apical annuli protein shown in hours of 3-indolacetic acid (IAA) auxin treatment observed by anti-V5 western blots. Histone H3 serves as a loading control, and molecular weight markers in kDa are shown. (**B**) Plaque assays in host cell monolayers showing plaque development over 8 days in knockdown cell lines for each apical annuli protein without (control) or with IAA-induced protein knockdown. (**C**) Replication states of *T. gondii* parasitophorous vacuoles 24 hr post invasion scored according to parasite number per vacuole. Each protein knockdown cell line is assayed either without (control) or with IAA-induced protein knockdown. Significant statistical differences between vacuole types are indicated by p-values *<0.05; **<0.01; ***<0.001, error bars = standard deviation.

The online version of this article includes the following source data and figure supplement(s) for figure 5:

**Figure 5-source data 1**

**Figure supplement 1.** Western blot and PCR analysis validating correct epitope-tag integration.

**Figure 5-figure supplement 1-source data 1**

---

*Dubois and Soldati-Favre, 2019*). Dense granule contents, however, are secreted post invasion and the site of their release from the parasite has been unknown. Our data for SNARE proteins at the apical annuli offers these locations as possible points for dense granule docking and exocytosis. To test if dense granule protein secretion is perturbed when either the apical annuli SNAREs or *Tg*LMBD3 are depleted, we assayed for GRA1, GRA2, and GRA5 secretion in our knockdown cell lines. Secreted GRA5 is delivered to the parasitophorous vacuolar membrane. IFAs for GRA5 were performed with digitonin cell permeabilisation that results in only limited disruption of the parasite plasma membrane. This enabled the secreted GRA5 to be preferentially detected (non-secreted GRAs will additionally occur in dense granules within the parasites). All four protein knockdowns showed reduced GRA5 delivered to the parasitophorous vacuole membrane (*Figure 6—figure supplement 1*). GRA1 and GRA2 are secreted into the parasitophorous vacuole space and upon fixation for IFAs both proteins typically show regions of accumulated signal between the parasites (*Figure 6B and C*, arrows). Depletion of *Tg*NPSN and *Tg*Syp7 showed clear reductions in this secreted signal (*Figure 6B and C*). When *Tg*LMBD3 was depleted a reduction in GRA1/2 secretion was less evident, and *Tg*StxPM depletion retarded vacuole development to the two-cell vacuole stages making visualisation of these secretion signals more difficult.

Non-secreted GRAs inside the parasites were evident in the GRA1 and GRA2 IFAs where Triton X-100 permeabilisation enabled antibody access to the dense granules. When the secreted GRA1/2 signal was reduced upon apical annuli protein knockdown, the GRA signal seen within the parasite was apparently increased compared to the control (*Figure 6B and C*). No change was evident in either microneme or rhoptry number or staining intensity when any of the four proteins were depleted

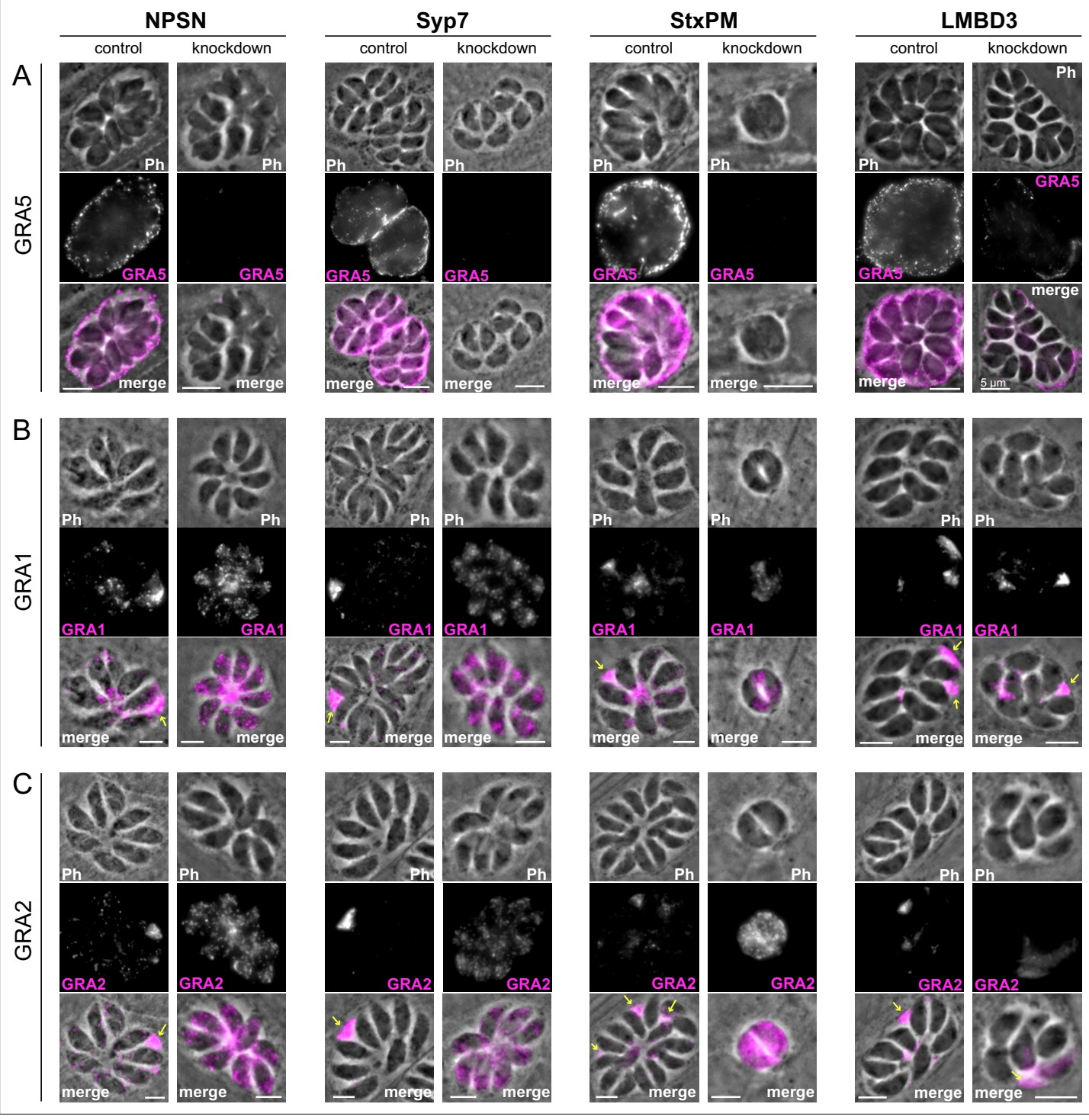

**Figure 6.** Secretion of dense granule proteins into the parasitophorous vacuole is inhibited when apical annuli membrane proteins are knocked down. Wide-field immunofluorescence assays of (**A**) GRA5, (**B**) GRA1, and (**C**) GRA2 without (control) or with auxin-induced protein knockdown for the four apical annuli membrane proteins. Assays for GRA5 were performed using digitonin permeabilisation to preferentially stain the secreted protein only (see *Figure 6—figure supplement 1* for further examples). Assays for GRA1 and GRA2 were performed with Triton X-100 permeabilisation to visualise the non-secreted dense granules as well as the secreted GRAs. Scale bar = 5 μm.

The online version of this article includes the following figure supplement(s) for figure 6:

**Figure supplement 1.** Wide-field images of parasitophorous vacuoles stained for GRA5 after depletion of the four apical annuli proteins.

**Figure supplement 2.** Abundance and distribution of rhoptry and microneme proteins are unaffected when apical annuli membrane proteins are knocked down.

(*Figure 6—figure supplement 2*). Hence, this elevation in dense granule number with apical annuli disruption is specific to this class of exocytic compartments. To quantify the changes in protein levels within the parasites when apical annuli function was perturbed, and to ask what other proteins are elevated within the parasite in these conditions, we performed quantitative shotgun proteomics on whole parasites grown in host cells for 24 hr with or without auxin treatment for three cell lines: *Tg*LMBD3-mAID-3xV5, mAID-3xV5-*Tg*NPSN, and mAID-3xV5-*Tg*Syp7 (mAID-3xV5-*Tg*StxPM was excluded from this analysis because of the risk of secondary effects of the severe growth phenotype with *Tg*StxPM depletion, and the challenge of harvesting adequate cell material from these poorly replicating cells). We predicted that many GRAs would show increased abundance in the parasite if dense granule exocytosis was inhibited. Protein abundance changes were visualised as volcano plots. For all three apical annuli protein depletions, dense granule proteins overwhelmingly dominated the proteins of increased abundance in the cells (*Figure 7*). For *Tg*LMBD3 depletion, the GRAs were almost exclusively the proteins whose increase was significant. Even the changes in GRA abundances that were assessed as non-significant in all three knockdowns at this timepoint were strongly skewed towards being increased. These data provide further strong evidence that perturbation of apical annuli function results in reduced secretion of dense granule proteins.

## Discussion

The *Toxoplasma* apical annuli have been previously described as features of the cell's IMC, beneath and separate from the plasma membrane. Here, we show that the annuli structures are relevant to the plasma membrane with four integral plasma membrane proteins at these sites including one that is exposed at the cell surface. We show that the annuli occur at small gaps in the IMC where three alveolar vesicles meet (the apical cap vesicle and two lateral vesicles) providing direct access of the cytosol to the plasma membrane. One of the apical annuli membrane proteins, *Tg*LMBD3, is a polytopic protein conserved throughout Apicomplexa and most eukaryotes. The other three represent Q SNARE proteins, containing Qa, Qb, and Qc SNARE motifs respectively, which are tethered by C-terminal transmembrane domains as is their conventional and ancestral state (SNARE protein fusions, and alternative membrane tethering mechanisms via post-translational added moieties, occur in many systems as derived states) (*Jahn and Scheller, 2006*; *Neveu et al., 2020*). Depletion of all four of these proteins affects dense granule secretion firmly implicating the apical annuli as the site of dense granule docking and membrane fusion.

The LMBD3 protein presents the first identified apical annuli protein that could interact with extracellular ligands and molecules. The functions of this group of proteins, however, are not well understood (*Redl and Habeler, 2022*). The four clear LMBD protein orthogroups found throughout eukaryotes suggest that the last eukaryotic common ancestor contained multiple of these proteins. While some eukaryotes, e.g., *Dictyostelium*, have retained genes for all four (*Kelsey et al., 2012*), most major eukaryotic groups have lost one or more of these paralogues. The first identified protein of this family, LMBR1, was associated with polydactyly and limb malformations in vertebrates, but its molecular role is still unclear (*Gyimesi and Hediger, 2022*). Other known members of the family are: LMBR1-like (LMBR1L, also known as LIMR), which interacts with lipocalins and other ligands involved in signalling cascades (Wnt/b-catenin and Nf-kB pathways); LMBR1 domain-containing protein 1 (LMBD1), which interacts with the lysosomal cobalamin transporter ABCD4; and LMBD2, which is a G-protein-coupled receptor-associated regulator of β2-adrenoceptor signalling (*Gyimesi and Hediger, 2022*). While *Tg*LMBD3 is in a different orthogroup to any of these studied paralogues, their common involvement in signalling events suggests that this protein of the apical annuli could contribute to the regulation of dense granule secretion, potentially in response to some extracellular stimulus.

SNARE proteins are classically responsible for driving membrane fusion events and our data strongly implicate the roles of *Tg*StxPM, *Tg*NPSN, and *Tg*Syp7 with dense granule fusion with the plasma membrane at the apical annuli. In other eukaryotes these SNARE orthologues all operate at the plasma membrane, and there is no evidence of duplication and/or specialisation of these SNAREs in Apicomplexa (*Dacks and Doolittle, 2002*; *Klinger et al., 2022*; *Venkatesh et al., 2017*). Vesicle fusion to a target membrane is driven through the regulated formation of the SNARE complex, a tight four-helix bundle of the Qa, Qb, and Qc SNARE motifs with an R SNARE motif-containing protein (*Jahn and Scheller, 2006*; *Neveu et al., 2020*). The Q SNAREs are anchored to the target membrane, and this is consistent with our observation of all three SNAREs being stably associated at the apical

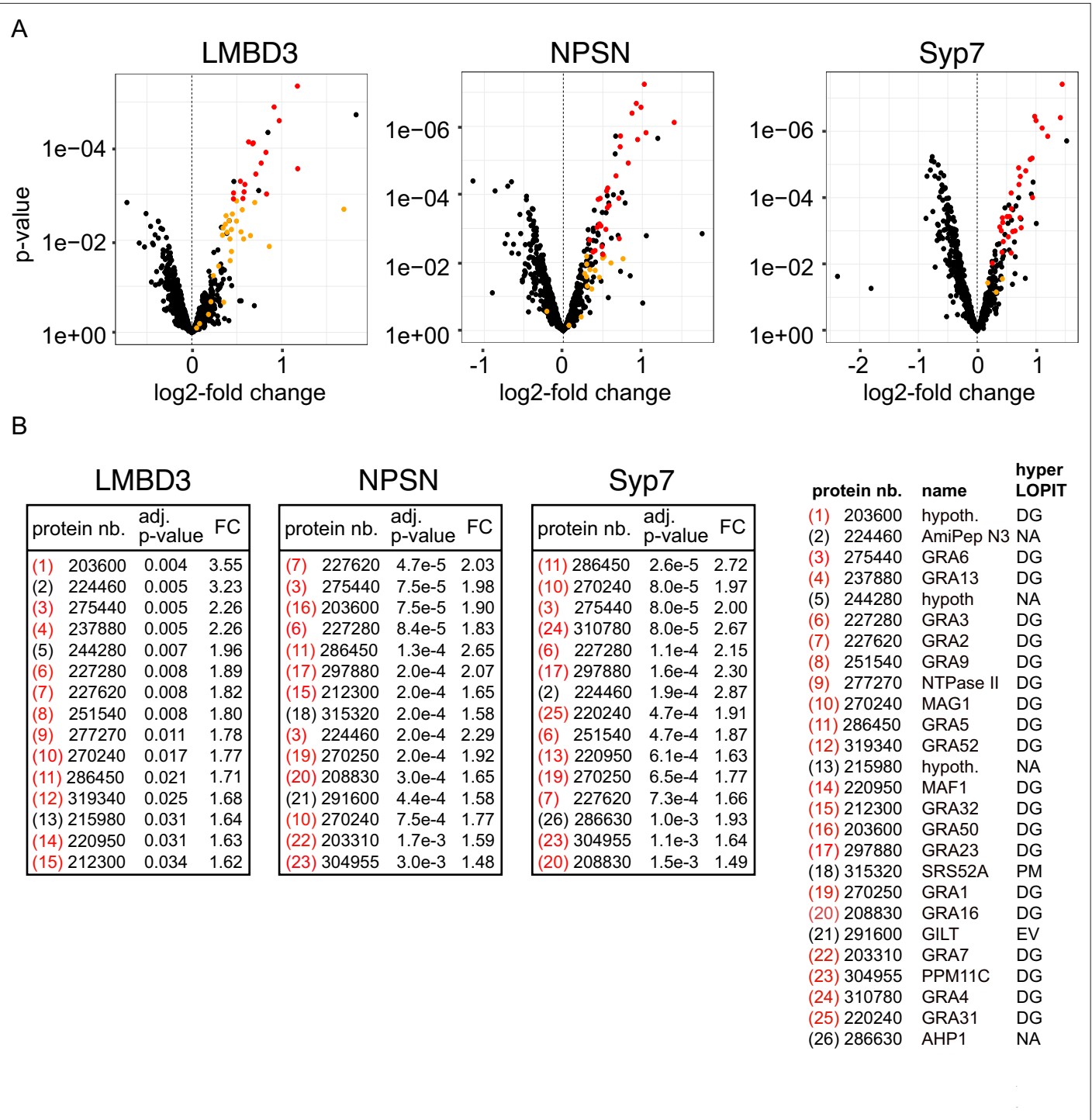

**Figure 7.** Knockdown of apical annuli membrane proteins results in accumulation of dense granule proteins in the parasite. (**A**) Volcano plots showing the changes of abundance of cell proteins with three apical annuli proteins depleted over 24 hr of auxin treatment compared to untreated controls (N=3). Black dots represent measures for individual cell proteins other than dense granule proteins, red and orange dots indicate dense granule proteins assigned by hyperLOPIT (red=statistically significant changes, orange=non-significant changes, using adjusted p-values) (***Barylyuk et al., 2020***). (**B**) The 15 most significantly increased proteins with each apical annuli protein knockdown amount to 26 common proteins randomly sampled by the shotgun proteomics, all but 6 of which are known dense granule proteins. Red bracketed numbers indicate GRAs, black bracketed numbers non-GRAs. FC, fold change; protein nb., VEuPathdb gene identifiers; hypoth., hypothetical protein; DG, dense granule; EV, endomembrane vesicles; PM, plasma membrane; NA, not assigned a location by hyperLOPIT. See ***Supplementary file 2*** for full quantitative proteomics data.

annuli. Moreover, our hyperLOPIT spatial proteomic data assigned all three as integral proteins of the plasma membrane, so it is unlikely that these SNAREs mediate any direct interaction with the IMC alveolar vesicle membranes despite their proximity (*Barylyuk et al., 2020*). A parallel study by *Fu et al., 2023*, also identified that these three *Toxoplasma* SNAREs occur at the apical annuli. They proposed alternative names: *Tg*Syntaxin-1 for *Tg*StxPM, *Tg*Syntaxin-21 for *Tg*NPSN, and *Tg*Syntaxin-20 for *Tg*Syp7. The 'Syntaxin' name is classically used only for Qa domain-containing SNAREs, and historically for animal orthogroups that function specifically in nerve cells. Furthermore, Syntaxin-1 to -4 are the names for specific vertebrate Qa paralogues that do not occur outside these animals. We, therefore, advocate use of the existing SNARE orthologue names used in this report as the most informative of origin and function.

It is curious that the knockdown phenotypes of the three SNAREs differ, with *Tg*StxPM showing the fastest arrest of parasite replication. These differences might be due to differing levels of reduction of dense granule secretion in each SNARE knockdown, and this could be due to either partial compensation from other cell Q SNAREs, or some residual functional protein in the knockdowns. We note also that *Tg*StxPM and *Tg*Syp7 show evidence of a second location within the cell. Some SNARE proteins are known to be able to participate in multiple membrane fusion events at different parts of the cell and with different SNARE partners (*Jahn and Scheller, 2006*), so a second (or more) role for these SNAREs might also explain their different growth phenotypes upon knockdown. In none of our SNARE knockdowns did we see obvious changes to the biogenesis of two other post-Golgi compartments, rhoptries, or micronemes, including differences in their total protein abundances, so these SNAREs do not appear to perform essential roles in these processes. Rather, the growth retardation with apical annuli protein depletion, particularly with *Tg*LMBD3 and *Tg*NPSN which are exclusive to these sites, is consistent with the reduced delivery of dense granule proteins into the host where they are required in both the parasitophorous vacuole membrane for harvesting host materials for the parasites' nutrition, and the host cytosol to abrogate host defence mechanisms (*Griffith et al., 2022*). *Fu et al., 2023*, reported no change in dense granule protein secretion in their experiments, however, they induced the SNARE depletions only after parasites had invaded the host cells. Dense granule exocytosis is known to occur very rapidly after invasion (*Dubremetz et al., 1993*; *Sibley et al., 1995*), so it is likely that their delay to SNARE depletion until well after invasion explains them not detecting the effect on this process.

Vesicle fusion typically requires an R SNARE that is anchored in the vesicle membrane, but we currently have not identified a candidate for the dense granules. However, other molecules contribute to the trafficking and delivery of vesicles to their sites of fusion (*Koike and Jahn, 2022*), and Rab11A has been identified to play this role for dense granules. Rab11A was shown to collocate with dense granules and follow their active movement along cytoskeletal fibres (*Venugopal et al., 2020*). The loss of function of Rab11A blocks dense granule protein secretion (*Venugopal et al., 2020*). Accompanying this block, an increase in dense granule numbers in the parasite's cytoplasm was seen, similar to what we saw with the SNARE knockdown by both microscopy and quantitative proteomics. Furthermore, knockdown of the apical annuli SNAREs results in Rab11A accumulating at the apical annuli (*Fu et al., 2023*), consistent with Rab11A being required for dense granule delivery and the SNAREs mediating subsequent vesicle fusion. While we have not observed dense granules in the act of docking and fusion at the apical annuli, this transient event is likely fast and difficult to capture. Unlike micronemes and rhoptries that tend to cluster towards the cell's apex and their sites of secretion, dense granules are uniformly scattered throughout the cell and are highly dynamic trafficking up and down actin networks (*Heaslip et al., 2016*; *Paredes-Santos et al., 2012*). This motility likely provides frequent access to their sites of secretion while avoiding crowding by these additional organelles in the relative confines of the cell's apical end.

The evidence of a second bespoke structure for vesicle exocytosis within the elaborate apicomplexan pellicle of *Toxoplasma* tachyzoites raises the question, why is the apical complex not sufficient? Both micronemes and rhoptries are known to pass through the conoid and exocytose at the apical plasma membrane via dedicated docking and fusion machinery (*Aquilini et al., 2021*; *Dubois and Soldati-Favre, 2019*; *Giuliano et al., 2023*; *Suarez et al., 2019*). Could dense granules not use this same site? The outcomes of each type of organelle secretion might be the reason why not. Microneme secretion from parasites within their host cells triggers parasite egress through permeabilisation of the host membranes prior to motile escape (*Roiko et al., 2014*). Dense granule secretion, on the other

hand, is required to actively maintain a stable intracellular host environment in which the parasite is nourished for replication (**Griffith et al., 2022**). The incompatibility of these two processes might, therefore, have driven their physical separation to avoid any potential 'leaky' or mistimed secretion from the wrong compartment (**Cova et al., 2022**; **Dubois and Soldati-Favre, 2019**).

With the discovery of the site of dense granule secretion, the regulation of this critical process can now come under closer scrutiny. Dense granule protein secretion has previously been described as constitutive or unregulated, however, it was subsequently observed to be negatively regulated by $Ca^{2+}$ in a reciprocal manner to the positive $Ca^{2+}$-driven exocytosis of micronemes (**Katris et al., 2019**). Centrin2 forms a ring at the inner side of the apical annuli, and $Ca^{2+}$-mediated constriction of centrin fibres might contribute to the closure of the annuli and inaccessibility of the plasma membrane to vesicles or docking machinery. Several other ring-forming components of the inner apical annuli structures contain cyclic nucleotide-binding domains, multiple phosphorylation sites, and other putative domains (rabaptin, gametogenetin) that could participate in vesicle trafficking and docking (**Engelberg et al., 2020**). Both $Ca^{2+}$ and cyclic nucleotides function as important messenger molecules that drive complex phospho-signalling cascades in apicomplexans that are at the heart of their invasion and proliferation programmes (**Bisio and Soldati-Favre, 2019**; **Uboldi et al., 2018**). These features of apical annuli proteins are, therefore, consistent with apical annuli being wired into these regulatory networks. It is possible that $Tg$LMBD3 might play a further role in regulation potentially responding to external cues. Dense granules are believed to contain upwards of 120 different cargo proteins in *T. gondii* (**Barylyuk et al., 2020**) so the coordination of release of this complex cargo into the host seems likely to be under strong selective constraints.

Where, then, did the apical annuli come from, and in what other related organisms might they occur? *Toxoplasma* and its very close relatives (*Hammondia, Neospora*) are relatively unusual in maintaining their complex cell pellicle throughout their asexual replication cycle (**Gubbels et al., 2021**; **Sheffield and Melton, 1968**; **Striepen et al., 2007**). While many other apicomplexans differentiate to lose the IMC as they feed and grow within their host's cells, *Toxoplasma* remains fully invasion-ready at all stages of growth. This maintenance of IMC, micronemes, rhoptries, and the apical complex, concurrent with the secretion of dense granule proteins, might necessitate the presence of apical annuli as alternative secretion points. *Eimeria* spp. on the other hand do lose their IMC after sporozoites invade the host's gut epithelial cells, and yet, freeze fracture studies of *Eimeria* sporozoite surfaces show apical annuli that look very similar to the structures in *Toxoplasma* (**Dubremetz and Torpier, 1978**) and apical annuli proteins AAP1–5 are all present in *Eimeria* (**Engelberg et al., 2020**). The presence of these structures in *Eimeria* might be because when sporozoites invade their hosts, there is a delay of multiple hours before the IMC is lost (**Pacheco et al., 1975**). The need for rapid post-invasion secretion of proteins that establish a safe and productive host environment, including defence against host immune attack, might therefore require apical annuli as secretion points even before the IMC and apical complex are dismantled. In turn, many other apicomplexans might require such secretion points to account for the fast pace of events from invasion to host modification that might need to precede IMC disassembly. Moreover, the delivery of plasma membrane carrier proteins and surface GPI-tethered molecules have implicated Rab11A, the apical annuli SNAREs and, therefore, the apical annuli in these further secretory pathways (**Fu et al., 2023**; **Venugopal et al., 2020**). Other apicomplexans might similarly have need for such secretion even in their invasion-ready cell forms. While some of the *Toxoplasma* apical annuli proteins appear more restricted to the Coccidia, this might primarily reflect the fast evolution of these structural proteins, many of which include coiled-coil domains (**Engelberg et al., 2020**). $Tg$LMBD3, on the other hand, is universally present in apicomplexans and their close relatives including the dinoflagellates that also have clear apical complexes with microneme and rhoptry-type secretory organelles by ultrastructural studies (**Dos Santos Pacheco et al., 2020**). It is possible that a much wider presence of apical annuli has been overlooked owing to their structure being more inconspicuous to traditional microscopy methods.

## Methods
### Growth and generation of transgenic *T. gondii*
*T. gondii* tachyzoites from the strain RH and derived strains, including RH Δku80/TATi (**Sheiner et al., 2011**), were maintained at 37°C with $pCO_2$ of 10% growing in human foreskin fibroblasts (HFFs)

cultured in Dulbecco's Modified Eagle Medium supplemented with 1% heat-inactivated fetal bovine serum, 10 unit ml$^{-1}$ penicillin, and 10 µg/ml streptomycin, as described elsewhere (*Roos et al., 1994*). Reporter protein-tagging of gene loci with reporters 6xHA, 3xV5, and eGFP was done according to our previous work (*Barylyuk et al., 2020*). When appropriate for selection, chloramphenicol was used at 20 µM, and pyrimethamine at 1 µM. The eGFP-Centrin2 cell line was selected by fluorescence activated cell sorting for eGFP. For protein function tests by gene knockdowns, the mAID cassette was fused to either the C- or N-terminus in Tir1 transgenic line using the same strategy as for reporter protein-tagging (*Koreny et al., 2023*). Proteins of interest were depleted with the addition of the auxin, IAA, at a final concentration of 500 µM. Oligonucleotides used for all gene modifications are shown in *Supplementary file 3*.

## Immunofluorescence microscopy

*T. gondii*-infected HFF monolayers grown on glass coverslips were fixed with 2% formaldehyde at room temperature for 20 min and permeabilised with 0.2% Triton X-100 for 10 min, except for GRA5 detection where 0.002% digitonin was used instead to selectively detect secreted GRA5. Blocking was done with 20% FBS for 1 hr, and the coverslips were then incubated with a primary antibody for 1 hr, washed in blocking buffer, followed by 1 hr incubation with a secondary antibody. Coverslips were mounted using ProLong Diamond Antifade Mountant (Thermo Fisher Scientific, MA, USA). Images were acquired using a Nikon Eclipse Ti wide-field microscope with a Nikon objective lens (Plan APO, 100×/1.45 oil) and a Hamamatsu C11440, ORCA Flash 4.0 camera. 3D-SIM was implemented on a DeltaVision OMX V4 Blaze (GE Healthcare, Issaquah, CA, USA) with samples prepared as for wide-field IFA microscopy with the exception that High Precision coverslips (Marienfeld Superior, No1.5H with a thickness of 170 µm±5 µm) were used in cell culture, and Vectashield (Vector Laboratories, Burlingame, CA, USA) was used as mounting reagent. Samples were excited using 405, 488, and 594 nm lasers and imaged with a 60× oil immersion lens (1.42 NA). The 3D-SIM images were reconstructed in softWoRx software version 6.1.3 (Applied Precision). All fluorescence images were processed using ImageJ software (http://rsbweb.nih.gov./ij/). The antibodies used, source, and the relevant concentrations are described in *Supplementary file 4*.

## Phylogenetic analyses

For the phylogenetic analyses, sequences were aligned using Mafft v7.407 with the L-INS-i algorithm (*Katoh and Standley, 2013*). Alignments were edited manually using Jalview (https://www.jalview.org). Maximum likelihood trees were calculated using PhyML-3.1 with bootstrap (1000 iterations) or SH-like aLRT branch supports (*Guindon et al., 2010*).

## Plaque assay

To test lytic cycle competence of knockdown cell lines by plaque formation in HFF monolayers, 200 freshly lysed parasites were added to six-well plates containing HFF monolayers. IAA was added to induce the gene knockdown, or an equivalent volume of ethanol added for uninduced controls. After 8 days of growth, flasks were aspirated, washed once with PBS, fixed with 5 ml of 100% methanol for 5 min, and stained with 5 ml of 1% crystal violet solution for 15 min. After staining, the crystal violet solution was removed, and the flasks were washed three times with PBS, dried, and imaged.

## Replication assay

The parasites were pre-treated with IAA (*Tg*LMBD3: 12 hr, *Tg*NPSN: 6 hr, *Tg*StxPM: 3 hr, *Tg*Syp7: 6 hr) or an equivalent volume of ethanol for the uninduced control before egress from the host cell to deplete the proteins of interest. Intracellular tachyzoites were then harvested through needle-passage using a 27 G hypodermic syringe needle and seeded on the HFF monolayer growing on coverslips in six-well plates. After 2 hr, uninvaded parasites were removed by washing and the invaded parasites were allowed to grow for further 24 hr with auxin or ethanol, followed by fixation for 20 min with 2% paraformaldehyde. Coverslips were then imaged using a Nikon Eclipse Ti microscope with a Nikon objective lens (Plan APO, 100×/1.45 oil), and a Hamamatsu C11440, ORCA Flash 4.0 camera. In haphazardly selected fields, the number of parasites per parasitophorous vacuole was scored. A minimum of 200 parasitophorous vacuoles was scored for each of the three biological replicates. p-Values were calculated with multiple t-tests and corrected for multiple comparisons using the

Holm-Sidak method in GraphPad Prism, v9 (GraphPad, CA, USA). A p-value of <0.05 was considered as significant.

## Trypsin shaving assay

The trypsin shaving assay was adapted from *Jia et al., 2017*. Fresh tachyzoite pellets were re-suspended in 0.1% trypsin/EDTA and incubated at 37°C for 1, 2, and 4 hr, and the control was treated with PBS only. After the incubation, samples were spun at 3000×*g* for 8 min, and supernatants were removed. Pellets were re-suspended in 1× Nupage LDS Sample buffer with either 50 mM DTT (followed by heated at 75°C for 10 min) or with 10 mM Tris(2-carboxyethyl) phosphine hydrochloride (Sigma) (followed by incubation at room temperature for 2 hr). Western blots were performed using rat anti-HA, mouse anti-Sag1, rabbit anti-GAP40, rabbit anti-PRF, and rabbit anti-TOM40 antibodies.

## Proteomics

### BioID sample preparation

For the proximity biotinylation assay, we generated a *T. gondii* cell line (in parental line RH Δku80) by endogenous tagging of the *Tg*LMBD3 locus with the in-frame coding sequence for the promiscuous bacterial biotin ligase, BirA*. The parental cell line was used as a negative control in biotin treatments. We followed the previously published BioID protocols (*Chen et al., 2017*; *Koreny et al., 2021*). Briefly, the parasites were grown in three biological replicates in elevated biotin concentration (150 µM) for 24 hr prior to egress, separated from the host cell debris and washed 5× in phosphate-buffered saline. The cell pellets were lysed in RIPA buffer by sonication and the lysates containing ~5 mg of total protein were incubated with 250 µl of unsettled Pierce Streptavidin Magnetic Beads (Thermo Fisher: 88817) overnight at 4°C with gentle agitation. The beads were then sequentially treated as follows: washed 3× in RIPA, 1× in 2 M urea and 100 mM triethylammonium bicarbonate (TEAB; Sigma); reduced in 10 mM DTT and 100 mM TEAB for 30 min at 56°C; alkylated in 55 mM iodoacetamide 100 mM TEAB for 45 min at room temperature in the dark; and washed in 10 mM DTT 100 mM TEAB, followed by 2×15 min in 100 mM TEAB with gentle agitation. The peptides were digested on the beads for 1 hr at 37°C incubation in 1 µg of trypsin dissolved in 100 mM TEAB, followed by an overnight 37°C incubation after adding an extra 1 µg of trypsin.

### Whole-cell quantitative proteomics sample preparation

Intracellular tachyzoites were pre-treated with IAA or with the vehicle (EtOH) prior to egress to deplete annuli proteins individually as for the replication assays, in three biological replicates for each cell line (*Tg*LMBD3: 12 hr, *Tg*NPSN: 6 hr, *Tg*Syp7: 6 hr). These were then harvested through needle-passage as before, seeded onto T175 flasks, and allowed to proliferate for further 24 hr with or without continued IAA treatment. Approximately 40 million parasites were then harvested by needle-passage into Endo buffer (44.7 mM $K_2SO_4$, 10 mM $MgSO_4$, 106 mM sucrose, 5 mM glucose, 20 mM Tris-$H_2SO_4$, 3.5 mg/ml BSA, pH 8.2) to mimic intracellular conditions. Host cell debris was removed by filtration through a 3 µm filter and tachyzoites were harvested by centrifugation at 1500×*g* for 10 min. Pellets were lysed in 8 M urea, prepared in 20 mM HEPES buffer, and sonicated for 5 cycles (50 s ON, 50 s OFF). Samples were reduced by the addition of DTT to a final concentration of 5 mM and incubated for 30 min at 37°C. Samples were then alkylated by the addition of iodoacetamide to a final concentration of 15 mM followed by incubation in the dark for 30 min. Proteins were then digested by the addition of trypsin/LysC mix of 25:1 protein:protease ratio (wt/wt) and incubation for 4 hr at 37°C. The reaction was then diluted eightfold or greater by adding 20 mM HEPES, pH 8 to reduce the concentration of urea to 1 M, and incubation at 37°C was continued overnight. Peptide digestion was terminated with the addition of trifluoracetic acid to a final concentration of 1%. Particulate material was pelleted by centrifugation at 21,000×*g*, 4°C, 10 min, and the supernatant was recovered for peptide desalting using Pierce Peptide Desalting Spin Column (Thermo Fisher: 89851) as described by the manufacturer. Peptide concentration was measured using the Pierce Quantitative Fluorometric Peptide Assay (Thermo Fisher: 23290) according to the manufacturer's instructions. From each sample, a volume containing 17 µg of peptide was dried in a vacuum centrifuge (SpeedVac SPD 1030), then re-suspended in 100 µl of 100 mM TEAB solution, pH 8.5.

## TMT-labelling and liquid chromatography and tandem mass spectrometry

TMT-labelling was done using either a TMT10plex isobaric tagging reagent (Thermo Fisher: 90110) for BioID samples or TMTpro 16 plex Label Reagent Set 1×5 mg (Thermo Fisher: A44520) for whole-cell protein quantitation. Each TMT reagent vial containing 0.5 mg of the labelling reagents was brought to room temperature and dissolved in 40 µl of LCMS-grade acetonitrile immediately before use. The TMT reagents were then split to two sets and 20 µl of the TMT reagents were added to each peptide sample. After incubating for 1 hr at room temperature, 5 µl of 5% hydroxylamine (vol/vol) was added to each sample, followed by incubation for 15 min to quench the reaction. The TMT-labelled fractions were combined and dried in a vacuum centrifuge (SpeedVac SPD 1030) at 4°C.

LCMS analyses were carried out on an Orbitrap Fusion Lumos Tribrid mass spectrometer coupled online with a Dionex Ultimate 3000 RSLCnano system (Thermo Fisher Scientific) as previously described (*Barylyuk et al., 2020*). The XCalibur v3.0.63 software was used to control the instrument parameters and operation, and record and manage the raw data. The LCMS system was operated in the positive-ion data-dependent acquisition mode with the SPS-MS$^3$ acquisition method with a total run time of 120 min. The dried TMT10 plex-labelled peptide samples were resolubilised in an LC-MS sample loading solution (0.1% aqueous formic acid) at a concentration of approximately 1 µg/µl. Approximately 1 µg of the sample was loaded onto a micro-precolumn (C18 PepMap 100, 300 µm i.d.×5 mm, 5 µm particle size, 100 Å pore size, Thermo Fisher Scientific) with the sample loading solution for 3 min. Following the loading step, the valve was switched to the inject position, and the peptides were fractionated on an analytical Proxeon EASY-Spray column (PepMap, RSLC C18, 50 cm×75 µm i.d., 2 µm particle size, 100 Å pore size, Thermo Fisher Scientific) using a linear gradient of 2–40% (vol) acetonitrile in aqueous 0.1% formic acid applied at a flow rate of 300 nl/min for 95 min, followed by a wash step (70% acetonitrile in 0.1% aqueous formic acid for 5 min) and a re-equilibration step. Peptide ions were analysed in the Orbitrap at a resolution of 120,000 in an m/z range of 380–1500 with a maximum ion injection time of 50 ms and an AGC target of 4E5 (MS$^1$ scan). Precursor ions with the charge states of 2–7 and the intensity above 5000 were isolated in the quadrupole set to 0.7 m/z transmission window and fragmented in the linear ion trap via collision-induced dissociation at a 35% normalised collision energy, a maximum ion accumulation time of 50 ms, and an AGC target of 1E4 (MS$^2$ scan). The selected and fragmented precursors were dynamically excluded for 70 s. Synchronous precursor selection (SPS) was applied to co-isolate 10 MS$^2$-fragments in the linear ion trap with an isolation window of 0.7 m/z in the range of m/z 400–1200, excluding the parent ion and the TMT reporter ion series. The SPS precursors were activated at a normalised collision energy of 65% to induce fragmentation via high-collision energy dissociation. The product ions were measured in the Orbitrap at a resolution of 50,000 in a detection range of m/z 100–500 with a maximum ion injection time of 86 ms and an AGC of 5E4 (MS$^3$ scan).

## Raw LCMS data processing

The processing of BioID raw LCMS data for peptide and protein identification and quantification was performed with Proteome Discoverer v2.3 (Thermo Fisher Scientific). Raw mass spectra were filtered, converted to peak lists by Proteome Discoverer, and submitted to a database search using Mascot v2.6.2 search engine (Matrix Science) against the protein sequences of *Homo sapiens* (93,609 entries retrieved from UniProt on 09.04.2018), *Bos taurus* (24,148 entries retrieved from UniProt on 17.04.2017), and *T. gondii* strain ME49 (8,322 entries retrieved from ToxoDB.org release 36 on 19.02.2018) (*Amos et al., 2022*). Common contaminant proteins – e.g., human keratins, bovine serum albumin, porcine trypsin – from the common Repository of Adventitious Proteins (cRAP, 123 entries, adapted from https://www.thegpm.org/crap/) were added to the database, as well as the sequence of the BirA* used to generate the BioID bait proteins by gene fusion. The precursor and fragment mass tolerances were set to 10 ppm and 0.8 Da, respectively. The enzyme was set to trypsin with up to two missed cleavages allowed. Carbamidomethylation of cysteine was set as a static modification. The dynamic modifications were set to TMT6plex at the peptide N-terminus and side chains of lysine, serine, and threonine, oxidation of methionine, deamidation of asparagine and glutamine, and biotinylation of the peptide N-terminus or lysine side chain. The false discovery rate (FDR) of peptide-to-spectrum matches (PSMs) was validated by Percolator v3.02.1 (*The et al., 2016*) and only high-confidence peptides (FDR threshold 1%) of a minimum length of six amino acid residues were used for protein identification. Strict parsimony was applied for protein grouping. TMT reporter ion

abundances were obtained in Proteome Discoverer using the most confident centroid method for peak integration with 20 p.p.m. tolerance window. The isotopic impurity correction as per the manufacturer's specification was applied. For protein quantification, PSMs with precursor co-isolation interference above 50% were discarded, and the TMT reporter ion abundances determined for unique (sequence found in proteins belonging to a single protein group) and razor (if sequence is shared by protein belonging to multiple protein groups, the quantification result is attributed to the best-associated Master Protein) peptides were summed.

The processing of whole-cell protein quantitation raw LCMS data for peptide and protein identification and quantification was performed with Proteome Discoverer PD v3.0 (Thermo Fisher Scientific) using an SPS MS$^3$ reporter ion-based quantification workflow. SequestHT was used as a search engine followed by an INFERYS rescoring node, checking spectra against a *T. gondii* ME49 proteome (ToxoDB-65), a swissprot human proteome, and a common contaminant database. Mass tolerances for peptide precursor and fragment ions were set to 10 ppm and 0.5 Da, respectively. Tryptic peptides were allowed to have two missed cleavage sites. Mass shifts were set up as either static modification for cysteine carbamidomethylation (+57.021 Da) and lysine TMTpro label (+304.207 Da) or as dynamic modification for methionine oxidation (+15.995 Da) and peptide N-terminal TMTpro label (304.207 Da). Percolator was used for FDR estimations with a fixed peptide target FDR of 1%. All PSMs up to delta Cn value of 0.05 were initially considered and only high-confidence peptides were retained. Contaminant proteins were removed. Quantification of peptides at the MS$^3$ level was performed using Most Confident Centroid as integration method (tolerance of 20 ppm). Reporter Abundance was based on signal-to-noise (S/N) values and corrected for isotopic impurities of TMT reagents according to the manufacturer's specifications (TMTpro 16plex LOT #VJ313476). Protein grouping was carried out applying the strict parsimony principle and proteins with high (q≤0.01) and medium (q≤0.05) confidence retained. The PD PSM output file was filtered and aggregated manually using R Bioconductor packages. PSMs were filtered for uniqueness (Number.of.Protein.Groups= 1), rank (Concatenated.Rank=1), ambiguity (Unambiguous+Selected), isolation interference (≤75%), average S/N (≥10), and SPS mass match percentage (≥70%) and aggregated to protein level applying robustSummary. Global protein abundances between samples were median aligned to account for slight variabilities due to peptide loading per TMT channel.

Raw LC-MS data and PD search results have been deposited to the ProteomeXchange Consortium (http://proteomecentral.proteomexchange.org) via the PRIDE partner repository (*Perez-Riverol et al., 2022*) with the dataset identifiers PXD034193 and PXD044588.

## Statistical analysis of proteomic data

BioID data analysis was performed with *R* v3.6.1 (*R Development Core Team, 2021*) using packages *tidyverse*v1.2.1 (*Wickham et al., 2019*) for data import, management, and manipulation, *Bioconductor* packages *MSnbase* v2.10.1 (*Gatto and Lilley, 2012*) or QFeatures (*Gatto and Vanderaa, 2023*) were used for managing quantitative proteomics data, *biobroom* v1.16.0 (*Bass et al., 2023*) for converting *Bioconductor* objects into *tidy data frames*, and *limma* v3.40.6 (*Ritchie et al., 2015*) for linear modelling and statistical data evaluation, as previously described (*Koreny et al., 2023*). The protein-level report generated by Proteome Discoverer was imported into R and filtered to remove non-*Toxoplasma* and low-confidence (protein FDR confidence level 'Low', q≥0.05). Only Master Proteins with a complete set of TMT abundance values across all replicates of the BioID bait and control samples were considered for the analysis. The protein abundance values in each biological sample were corrected for the total amount using normalisation factors derived from the abundances of two proteins, acetyl-CoA carboxylase ACC1 (TGME49_221320) and pyruvate carboxylase PC (TGME49_284190). Both proteins are highly expressed, endogenously biotinylated, and reside in the matrix of subcellular compartments, the apicoplast and mitochondrion for ACC1 and PC, respectively, where they are not accessible to the BirA*-fused BioID baits. Hence, these two proteins served as suitable internal standards. The normalised protein abundances were log2-transformed and fitted with a linear model in *limma*. In our experimental setup, we distributed the samples between three TMT10plex sets with one biological replicate of the BirA*-tagged and three replicates of the control cell lines per set. The mean protein abundances in these two groups of samples were modelled as a simple linear relationship with three coefficients: the intercept representing the reference protein abundance level (condition:control), the condition coefficient (condition:BirA*-tagged) representing

the difference in protein abundance between the two groups, and the batch coefficient accounting for the possible batch effect between three TMT10plex sets. The model tested the hypothesis that the mean protein abundances in both sample groups, the control and the BirA*-tagged, are equal, by testing that the second coefficient is equal to zero. If the condition parameter estimated by *limma* linear model was significantly different from zero, we concluded that the condition (presence of the BirA*-fused bait) had a significant effect on the protein abundance. Also, *limma* estimated the model parameters taking into account the relationship between protein average intensities and the variance (low-abundance proteins tend to have a greater variance) by empirical Bayesian shrinking of the standard errors towards the trend. This enabled a better control of false discoveries and outliers affording more robust identification of significantly enriched proteins. The resulting p-values were adjusted for multiple testing using the Benjamini-Hochberg method (FDR <1 %).

For whole-cell protein quantitation, differential protein abundance between control and treated cell lines for *Tg*LMBD3-mAID-3xV5, mAID-3xV5-*Tg*NPSN, and mAID-3xV5-*Tg*Syp7 was determined by implementing the Bioconductor package *limma*, using a mean reference model with the lmFit and eBayes functions. Proteins with an adjusted p-value <0.05 (Benjamini-Hochberg correction) after moderated t-tests were considered significant. The complete R markdown file used for the analysis is provided (*Supplementary file 5*).

## Acknowledgements

This work was supported by the Wellcome Trust, United Kingdom, Investigator award 214298/Z/18/Z, and Gordon and Betty Moore Foundation grant BGMF7872 (Doi:10.37807/GBMF9194). We thank VEuPathDB for their invaluable Informatics Resources, the Cambridge Centre for Proteomics for sample analysis, the flow cytometry facility from the School of the Biological Sciences, and Jan Pyrih and Brandon Mercado-Saavedra for useful discussions.

## Additional information

### Funding

| Funder | Grant reference number | Author |
| --- | --- | --- |
| Wellcome Trust | 214298_Z_18_Z | Sara Chelaghma<br>Huiling Ke<br>Konstantin Barylyuk<br>Ludek Koreny<br>Ross F Waller |
| Gordon and Betty Moore Foundation | 7872 | Thomas Krueger<br>Ross F Waller |

The funders had no role in study design, data collection and interpretation, or the decision to submit the work for publication. For the purpose of Open Access, the authors have applied a CC BY public copyright license to any Author Accepted Manuscript version arising from this submission.

### Author contributions

Sara Chelaghma, Huiling Ke, Formal analysis, Investigation, Methodology, Writing – review and editing; Konstantin Barylyuk, Thomas Krueger, Formal analysis, Methodology, Writing – review and editing; Ludek Koreny, Conceptualization, Supervision, Writing – original draft, Writing – review and editing; Ross F Waller, Conceptualization, Supervision, Funding acquisition, Investigation, Writing – original draft, Writing – review and editing

### Author ORCIDs

Sara Chelaghma http://orcid.org/0009-0004-1832-4406
Konstantin Barylyuk http://orcid.org/0000-0002-3580-0345
Thomas Krueger http://orcid.org/0000-0002-8132-8870
Ludek Koreny https://orcid.org/0000-0002-9979-3172
Ross F Waller https://orcid.org/0000-0001-6961-9344

Decision letter and Author response
Decision letter https://doi.org/10.7554/eLife.94201.sa1
Author response https://doi.org/10.7554/eLife.94201.sa2

## Additional files

### Supplementary files

• Supplementary file 1. BioID supplementary data file. Sheet 1: BioID_significant_changes: Columns show ToxoDB accession number, log2-fold change (logFC) with left and right confidence intervals (CI.L, CI.R), the average log2-abundance value of this protein across treatments and their replicates (AveExpr), the moderated t-statistics value (t), the raw p-value (P.Value), the adjusted p-value (adj.P.Val), and the log-odds that the protein is differentially abundant (B). Sheet 2: Normalised TMT intensity values for three LMBD3-BirA* biological replicates (RUN1-3) each with three parental control samples.

• Supplementary file 2. Quantitative proteomics supplementary data file. Sheets 1–3 give data for the three cell lines: TgLMBD3-mAID-3xV5, mAID-3xV5-TgNPSN, and mAID-3xV5-TgSyp7. Columns show ToxoDB accession number, number of peptides, log2-median-aligned protein abundances for all replicates in knockdown (KD_rep1-rep3) and control treatment (control_rep1-rep3), treatment means and standard deviation (SD), effect size (Cohen's D), statistical power for two-sided t-test at p=0.01, log2-fold change between treatment and control (logFC), the average log2-abundance value of this protein across treatments and their replicates (AveExpr), the moderated t-statistics value (t), the raw p-value (P.Value), the adjusted p-value (adj.P.Val), the log-odds that the protein is differentially abundant (B), the protein description (Description), and the TAGM-predicted subcellular location according to the ToxoLOPIT map of *Barylyuk et al., 2020*.

• Supplementary file 3. Primers and plasmids for genetic modifications.

• Supplementary file 4. Antibodies used for immunofluorescence assays (IFAs) and Western blots.

• Supplementary file 5. R markdown file of analytical workflow of quantitative proteomics data. The markdown file contains the pipeline from the peptide-to-spectrum match (PSM)-level input data obtained from Proteome Discoverer. It provides an overview on structure and quality of the raw data (chunk 1–5), explores missing data structure and protein coverage across experiments (chunk 6–7), aggregates the psm-level data to proteins (chunk 8), and creates the linear model fits using Limma Bayes algorithms (chunk 10). The last two chunks (11+12) create the output data files and Volcano plots submitted with this manuscript.

• MDAR checklist

### Data availability

Raw LC-MS data and PD search results have been deposited to the ProteomeXchange Consortium via the PRIDE partner repository with the dataset identifiers PXD034193 and PXD044588.

The following datasets were generated:

| Author(s) | Year | Dataset title | Dataset URL | Database and Identifier |
|---|---|---|---|---|
| Barylyuk K, Waller RF | 2023 | Protein composition of Toxoplasma endocytic and exocytic structures by proximity-dependent biotinylation | https://www.ebi.ac.uk/pride/archive/projects/PXD034193 | PRIDE, PXD034193 |
| Krueger T, Waller RF | 2023 | Apical annuli are specialised sites of post-invasion secretion of dense granules in *Toxoplasma* | https://www.ebi.ac.uk/pride/archive/projects/PXD044588 | PRIDE, PXD044588 |

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
