## [Editor Report]

This important study identifies a novel mechanism for protein secretion in the obligate human protozoan parasite *Toxoplasma gondii*. The authors utilize a range of advanced imaging, proteomic and genetic approaches to convincing demonstrate the mechanism by which dense granule proteins are delivered to the parasite plasma membrane. This study will be of broad interest to cell biologists and parasitologists working on membrane trafficking and virulence mechanisms.

---

## [Decision Letter]

[Editors' note: this paper was reviewed by Review Commons.]

---

## [Author Response]

General Statements [optional]

We have made a full revision of our report acting on and responding to all of the reviewers’ comments and suggestions (see below). In particular, we have included fuller comparison of our work to that of a parallel study (Fu et al., 2023) which also identified three SNARE proteins at the sites of secretion that we address. A major difference in their study is that they failed to identify that dense granule, which are major secretory organelles that drive post-invasion host-parasite interactions, secrete their ‘GRA’ proteins at these sites. Reviewers 2 and 3 mistakenly assume that Fu et al. did come to this conclusion, but this is clearly stated in their report to not be the case: ‘In our experiments, none of the SNAREs were shown to be related to the exocytosis of GRAs. Therefore, the mechanism that mediates exocytosis of GRAs at the plasma membrane remains to be elucidated.’ Fu et al. (2023). In our revision we provide a clear explanation for why their study was insensitive to this important process and discovery.

We have also included further examples of some images as supplemental data as requested by Reviewer #3, removed one discussion element that required speculation of SNARE-independent exocytosis of rhoptries and micronemes, and made some very minor corrections to the methods description of the proteomics.

Point-by-point description of the revisionsReviewer #1 (Evidence, reproducibility and clarity (Required)):Summary*Toxoplasma gondii* is an obligate intracellular parasite. Intracellular survival critical depends on secretory vesicles named dense granules. These vesicles are predicted to contain >100 different proteins that are released into PV, PV membrane and the host cell to control the parasites intracellular environment and host cell gene expression and immune response. How and where these vesicles are released from the parasite is a long-standing question in the field because *T. gondii*, and other apicomplexan parasites contained a complex pellicular cytoskeletal structure called the IMC which limits dense granule access to the plasma membrane. In this manuscript by Chelaghma, Ke and colleagues demonstrates for the first time that dense granules are secreted from the parasite at pore structures called the apical annuli. The authors used their previously generated HyperLOPIT data set and identified a plasma membrane protein that is specifically enriched at the apical annuli. Using BioID the authors then identify three SNARE proteins that also localize at the apical annuli. The localization of these proteins is determined using excellent super-resolution structured illumination microscopy. Conditional protein knockdowns for all four proteins were created and both proteomics and microscopy used to demonstrate a reduction in dense granule secretion in the absence of these proteins. Collectively, these data make new and substantial contributions to our understanding of mechanisms of dense granule secretion.Major comments:Overall, these data is convincing and well-described. The text is clear and well written. There are a few instances (see below) where the authors doesn't adequately describe the data or over state the strength of the results. These issues could all be addressed editorially or by process existing data.Comment 1.1The authors use proteomics and IFA to show that there is a reduction (rather than an inhibition of) in dense granule secretion. However, from the phase images in figure 5, the vacuoles of KD parasites look normal and so not have the phenotypes that one would expect after a significant reduction in dense granule secretion, such as the "bubble" phenotype described for GRA17 and GRA23 knockouts (Gold et al. 2015; PMID: 25974303). Authors should describe their findings in the context of the expected phenotypes based on the published literature. The statement on line 369-371 is too strong and should imply a reduction rather than an inhibition of dense granule secretion.

Authors’ response: It is difficult to compare our results to individual dense granule protein mutants described in the literature because such phenotypes are the result of the loss of only a single protein being exported to the host, whereas we are observing the effects of the reduction of secretion of up to 120+ different proteins. Furthermore, we agree with this reviewer that none of the protein knockdowns appear to completely prevent dense granule secretion, which we implied by ‘inhibition’, and this could be either due to incomplete knockdown of each of these proteins with some residue function, or some redundancy where other proteins can contribute to secretion. We have changed the statement flagged by this reviewer to: ‘Depletion of all four of these proteins affects dense granule secretion’ to avoid the interpretation of complete loss of function. We now further state that residual secretion may still occur and consider this in the light of possible reasons for this (Discussion, paragraph 4). In any case, none of these considerations change our conclusion that these proteins, at the site of the apical annuli, are implicated in dense granule secretion.

Comment 1.2The more severe phenotype observed in the AAQa iKD and the additional localizations of AAQa and AAQc suggests an additional role for these protein in protein trafficking that is supported by the authors data. In both AAQa and AAQc there appears to be an accumulation of GRA1 in a post-Golgi compartment and is less vesicular in appearance than the phenotype observed in the AAQb iKD parasites. Additionally, I disagree with the authors assessment that KD of these proteins does not effect microneme localization. In both AAQa and AAQc there appears to be increased number of micronemes at the basal end of the parasites compared with controls. Although this is not a direct focus of the authors papers, a description of these findings should be included in the results and Discussion sections.

Authors’ response: We have included a more complete discussion that considers the differences in phenotypes of the four mutants, including additional locations of two SNAREs, all of which is consistent with known SNARE biology (Discussion, fourth paragraph). These considerations, however, have no impact on our conclusions where all four proteins, including two that are exclusive to the apical annuli, have equivalent effects on dense granule exocytosis.

Concerning the effects on microneme and rhoptries of the different knockdowns, we have modified and limited our interpretation to overall IFA staining strength and protein organelle protein abundance by proteomics, where we see no differences. This addresses if there is a major post-Golgi trafficking defect that could affect biogenesis of all of micronemes, rhoptries and dense granules, for which we see no evidence. Whether there are subtle differences in the location of these organelles, which are known to show some variability, is beyond the scope or relevance to our central questions. Given that growth phenotypes are seen for all mutants, it is quite possible that secondary effects of retarded cells might present as some disorder within the cell, although we saw nothing conspicuous of this nature in many hundreds of examples observed.

Comment 1.3Presentation of the data in Figure 5. This figure contains images where the fluorescent dense granule signal is overlaid on phase images. However, in some cases (AAQb, AAQc, AAQa, GRA1 KD) the merged imaged looks like a straight merges of the two images, whereas in the rest of the images it looks like a thresholded fluorescent image is merge with phase. Authors need to process the images in consistent manner and provide a description of the image processing in the figure legend and Materials and methods.

Authors’ response: Thank you for this suggestion, we have now processed all of these merges the same way (ImageJ -> merge channels -> Composite Sum). While the merges are only intended to aid in aligning the fluorescence signal with the phase image, we agree that it is better to present them the same way.

Minor comments:Comment 1.4The discussion is overly long and could be shorted in some places. Lines 373 and 388 in particularly don't seems directly relevant to the manuscript.

Authors’ response: The paragraph identified by this reviewer considers the LMBD protein that is the first, and currently only, trans plasma membrane protein specific to the apical annuli that implies that this structure is exposed to the exterior of the cell. It is, therefore, of considerable significance to how we interpret the function and behaviour of these annular structures. We believe that it is very relevant to our study to consider what else is known about these relatively mysterious, but widely conserved, eukaryotic proteins, which is the subject of this paragraph. The other reviewers highlight the relevance of LMBD3 to the interpretation of this structure. This reviewer hasn’t identified any further superfluous discussion elements, and we believe that the current length is not excessive and is justified.

Comment 1.5Line 184 – Remove question mark from this sentence

Authors’ response: The question mark has been removed.

Comment 1.6Line 321. Should read Figure 7A, not figure 6A.

Authors’ response: Thank you, corrected.

Comment 1.7Line 139 – should read Figure 1B instead of 2C

Authors’ response: Thank you, corrected (although to 1C, which is in fact correct).

Comment 1.8Figure 3- Column labels for early, mid, or late endodyogeny would help with the clarity of this figure, especially for readings who are unfamiliar with the field.

Authors’ response: We have labelled the figure as suggested.

Comment 1.9Figure S2 – the letter n is missing from knockdown labels. And the number 3 from LMBD 3 is covering the word knockdown in the last panel.Authors’ response: Thank you, corrected.Reviewer #1 (Significance (Required)):The manuscript provides, for the first time, insight into the mechanism of dense granule secretion in Toxoplasma and identifies the sites on parasite pellicle where these vesicles can traverse the IMC to reach the plasma membrane. This is a significant conceptual advance in our understanding of this cellular vital process, one that is required for *T. gondii* intracellular survival. This paper would have broad interest from other research groups studying parasitology, secretion and protein trafficking.Reviewer #2 (Evidence, reproducibility and clarity (Required)):Summary:This manuscript reports on characterizing the function of the long-known apical annuli, which are pores embedded in the membrane skeleton of *Toxoplasma gondii*. Since their function has remained long elusive, this manuscript is a major breakthrough.Comment 2.1It is of note, however, that this breakthrough, using the same three SNAREs, was recently, in parallel, also reported by Fu et al. in PLoS Pathogens (PMID 36972314), which work is cited here. The additional novelty here is the finding of LMDB3 in the plasma membrane at the site of the annuli. This is a widely conserved protein for which little function is known except roles in signaling, The connection between LMDB3 and the SNAREs is through BioID, but they are preys quite far down the list. Furthermore, the function of LMDB3 is not explored here. As such, the additional advance compared to the Fu et al. report is limited. The function of the SNAREs in dense granule exocytosis is much more robustly done here through the proteomics data displaying an accumulation of DG proteins.

Authors’ response: While it is true that the discovery of the three SNAREs at the apical annuli was made and reported in parallel by Fu et al. (2023), a major difference in their conclusions is that they suggest that dense granules are not secreted at this site (this reviewer has mistakenly thought that this was their conclusion – “In our experiments, none of the SNAREs were shown to be related to the exocytosis of GRAs. Therefore, the mechanism that mediates exocytosis of GRAs at the plasma membrane remains to be elucidated.” Fu et al. (2023)). The failure of Fu et al. to detect this was almost certainly because they only tested for dense granule secretion defects by inducing depletion of the apical annuli SNAREs after the parasites had invaded the host cells. It is known that dense granule protein secretion happens rapidly in the initial moments after invasion, so apical annuli perturbation in their assay would have only occurred after these secretion events. We directly discuss this experimental difference in our revised discussion and how it accounts for their different conclusions (Discussion, fourth paragraph). We independently tested for this effect by quantitative proteomics which further supported our conclusions.

As this reviewer indicates, we additionally discovered that a protein (LMBD3) also spans the plasma membrane at these structures, and this implicates signalling or events at the cell surface. We show that this protein is also required for normal dense granule secretion. While we have not identified an explicit mechanistic role for LMBD3 in this process, such insight is also lacking for all LMBD proteins, including those in humans where they are implicated in disease. While we continue to pursue this interesting question of LMBD3 function, we are by no means alone in cell biology for these answers to be outstanding still.

Comment 2.2The presentation of the data is very clean and convincing, and the broader evolutionary context is well-presented as well. The discussion on whether maintaining the IMC during cell division is an innovation or ancestral is an open debate where the authors seem to come down on the side of innovation, but the evidence could go either way, so I would caution a bit more.

Authors’ response: We are puzzled by this reviewer’s comment because we do not make reference to the maintenance of the IMC during cell division in this evolutionary context — ancestral or a recent innovation. We describe the case of Toxoplasma and its close relatives maintaining the maternal IMC during division as ‘unusual’, not ancestral (second sentence of the last paragraph of the Discussion), and this is the only statement that we think might have elicited this query from the reviewer. But this does not imply what the ancestral state might have been which is not a subject of any of our considerations here.

Major comments:– Are the key conclusions convincing?Comment 2.3The identification of the three SNARE proteins through BioID is not very convincingly represented in Table S1. These SNAREs were not showing significant changes and were not detected universally across the three bio-reps, and thyn were also present in the controls. Although this does not diminish the message of the work, this appears to be quite Cherry-picked, while other top hits in the BioID were overlooked, e.g. Nd6 and Nd2 are right in the top ten, which have a demonstrated role in rhoptry exocytosis. This certainly piqued my interest, but is not even discussed.

Authors’ response: We have used BioID as a protein discovery strategy, not to directly measure protein proximity for which it is an imperfect measure for many technical reasons. Accordingly, discovered ‘candidates’ for proteins that might occur at the annuli were all independently verified by protein reporter tagging. We focused our efforts on discovering apical annuli plasma membrane-tethered proteins and, therefore, parsed our BioID data for those shown previously to be in the plasma membrane by LOPIT spatial proteomics (Barylyuk et al., 2020). It is true that the SNARE proteins were not favoured over many other proteins in the BioID signal, but their verified location at these sites justified our pursuit of them as new apical annuli proteins.

Other proteins, including the previously identified apical proteins Nd6 and Nd2 that are implicated in rhoptry secretion, similarly piqued our interest! But when we reporter-tagged them they were revealed as BioID false positives, consistent with published work on these proteins, and other ‘top hits’ included some other false positives. Table S1 is included as a further recourse for the field, but it only served as a first step in functional protein discovery in our study.

Comment 2.4TgAAQa, TgAAQb and TgAAQc were recently also reported to localize to the annuli by Fu et al. 2023 (PMID: 36972314; this report is even cited in this manuscript for Rab11a accumulation), who gave them different names: TgStx1, TgStx20, and TgStx21 (not in this order). I see no reason to adopt a new nomenclature here, which will be very confusing in the future literature. Please adopt the Stx names in this manuscript.

Authors’ response: We agree that where there is precedent in naming it is better to use the earliest used names. Naming of proteins is also best done to reflect orthologues found between species so that consistent names indicate common functions. The naming system proposed by Fu et al. for the Qa, Qb and Qc SNAREs unfortunately does not fulfil this second important criterion. They based their names on ‘Syntaxin’ which was first used for an animal SNARE of the nervous system that is almost exclusively used for Qa paralogues. Furthermore, in animals Stx1-4 are all vertebrate-specific Qa paralogues that have arisen only in this group. So, to name the Qa SNARE of Toxoplasma according to one of these animal-specific nerve proteins (Stx1) implies an evolutionary inheritance that is very unlikely (i.e., lateral gene transfer from an animal) and is unsupported by published phylogenies. Furthermore, Fu et al. also give the Qb and Qc SNAREs the animal Qa name ‘syntaxin’, and arbitrarily number them Stx21 and Stx20. So, while they have named these proteins first, we think that the names given provide confusing and misleading labels for these proteins.

We initially proposed a simpler system according to the location of the SNARE in Toxoplasma (AA = Apical Annuli) and the Q domain type (Qa, Qb, Qc), e.g., AAQa. But on reflection we propose using precedent and orthology and adopt the existing orthologue names as the most useful solution. Klinger et al. (2022) have resolved the phylogeny of the three Toxoplasma SNAREs, and they group with strong phylogenetic support with known eukaryote-wide orthogroups with previous names: Qa=StxPM (Syntaxin Plasma Membrane); Qb=NPSN (Novel Plant ‘Syntaxin’); and Qc=Syp7 (a Qc SNARE family originally thought to be specific to plants). These SNARE types are all known to operate at the plasma membrane, and accordingly the names TgStxPM, TgNPSN, and TgSyp7 would indicate their orthology and similar functional location known in other eukaryotes. We have justified this preferred naming system in the text of our report (Discussion, third paragraph), but making it clear which Fu et al. names correspond to these more universally consistent names so that these can be easily cross-referenced.

Comment 2.5No knock-down of LMBD3 is pursued: how would this impact SNARE distribution and/or other annuli proteins? The fitness score is very severe, -4.07, so this is somewhat puzzling. Lower comment is related. This could provide tantalizing insights in the architecture of the annuli, and/or their function as a secretory conduit.LMBD3 relative to the SNAREs is not explored: co-IPs or detergent extraction to see if they are all in a physically interacting complex. What keeps them together. Is LBCDR3 interfacing with any annuli proteins Cen2 is suggested through the image in Figure 2A, though there appears to be some separation in some images: AAP2, 3 and 5 were previously shown to have smaller diameters than Cen2 and therefore appear better positioned.

Authors’ response: LMBD3 knockdowns were pursued in so far as identifying that they also have a phenotype of reduced dense granule secretion as for the SNAREs, but it will indeed require further studies of this intriguing molecule to define its specific function. Our central questions of this study were what is the association of the apical annuli with respect to the IMC and plasma membrane, and what is the overall significance and function of these structures. These core questions have been answered in our study. The questions that this review raises here are further and logical questions specifically related to LMBD3 that we are now pursuing as an independent follow-on study.

– Should the authors qualify some of their claims as preliminary or speculative, or remove them altogether?Comment 2.6The discussion on whether maintaining the IMC during cell division is an innovation or ancestral is an open debate where the authors seem to come down on the side of innovation, but the evidence could go either way, so I would caution a bit more.

Authors’ response: This comment (2.2) is already made and addressed above.

– Would additional experiments be essential to support the claims of the paper? Request additional experiments only where necessary for the paper as it is, and do not ask authors to open new lines of experimentation.Comment 2.7The heavy focus on the LMBD3 in Figure 1 and the evolutionary discussion would warrant a more direct functional dissection. Either through an LMDB3 known-down, or its interface with the SNAREs or annuli more directly.

Authors’ response: This reviewer has not made it clear that further work on LMBD3 is necessary to support the conclusions of the paper or address the questions that we have asked, only that they would like to see more insight into LMBD3. We would also! But we do present knock-down studies and show that there are functional consequences for dense granule secretion. The question of if LMBD3 is involved in the maintenance of apical annuli structure and/or integrity is an interesting one, but a further question to those that we have presented in this first study. LMBD proteins have poorly characterised molecular functions throughout eukaryotes, and while we are also motivated to understand their role more, this has not proven a straightforward task in other systems also.

Comment 2.8The claim that the annuli are the conduits though which the dense granules travel to get exocytosis is not directly supported by any of the experiments as it is solely based on co-localization studies, not even direct interactions.

Authors’ response: We agree that we have not directly observed dense granules in the act of secretion at the apical annuli. Dense granules are known to be very mobile in the cell and traffic dynamically on actin networks. So, they do not accumulate at any one site, and their fusion and exocytosis is likely a rapid, transient event. Multiple lines of evidence for them pausing and fusing with the plasma membrane, while indirect, independently support this conclusion:

1)SNARE proteins restricted to the apical annuli in the plasma membrane are required for normal dense granule secretion2)When these SNAREs are depleted dense granule proteins accumulate in the parasite3)Rab11A is a further vesicle-tethering molecule that has been shown to be attached to dense granules and its mutation also leads to inhibition of dense granule proteins (Venugopal et al., 2020)4)When the apical annuli SNAREs are depleted Rab11A accumulates at the annuli (Fu et al., 2023)

Collectively, we believe that the claim that the apical annuli are the sites of dense granule secretion is very strongly supported, particularly by the very molecules that would be required for vesicle docking and fusing at these sites, and is justified to be noted in the title. We have, however, made it clear in our report now that these data are indirect and that dense granules are yet to be captured in the act of secreting their contents at these sites (Discussion, paragraph five).

Referees cross-commentingThe consolidating themes I see (and value) in the reviews:Comment 2.91. functional follow up of role of LMDB3

Authors’ response: This work is already part of a follow-up project.

Comment 2.102. adopt nomenclature of Fu et al., to avoid confusion in literature

Authors’ response: Please see our response to Comment 2.4

Comment 2.113. better integrate the findings in light of the Fu et al. publication throughout this manuscript

Authors’ response: We have further acknowledged and compared our findings to those of the parallel study of Fu et al. with additional text in the discussion.

Comment 2.124. no direct evidence of dense granules at annuli; attenuate the claims (in title etc), or include supportive data

Authors’ response: Please see our response to the equivalent Comment 2.8 above.

Reviewer #2 (Significance (Required)):– Describe the nature and significance of the advance (e.g. conceptual, technical, clinical) for the field.Comment 2.13The presented manuscript reports on a novel protein, LMBD3, embedded in the plasma membrane of *Toxoplasma gondii* at the site of the apical annuli, which are pores across the inner membrane complex (IMC) skeleton. This provides a novel, putative connection between the cytoplasm and plasma membrane, although this is not directly explored here. Through LMDB3 proximity biotinylation, three SNAREs are identified that were recently reported to be involved in dense granule exocytosis, which is is confirmed here through robust proteomic experiments.

Authors’ response: This reviewer has made an error here in stating that the parallel study of Fu et al. implicated the apical annuli SNAREs with dense granule exocytosis. See our response to Comment 2.1 where we describe why the experimental design used for Fu et al. was unlikely to test this question effectively.

– Place the work in the context of the existing literature (provide references, where appropriate).The annuli were first reported in 2006, and understanding of their proteomic composition has expanded over the years, however, a function has remained long elusive. This report, together with another parallel performed work, now uses three SNAREs, named TgAAQa, TgAAQb and TgAAQc in this report but previously named TgStx1, TgStx20, and TgStx21 (not in this orthologous order), localizing to the annuli as tool to assign the function of the annuli to exocytosis of the dense granules during intracellular parasite multiplication. The evolutionary context and concepts of the new findings are very well-embedded in the existing literature and insights.– State what audience might be interested in and influenced by the reported findings.The audience comprises people with a specific interest beyond apicomplexan biology, basically all Alveolates as they all share a similar membrane skeleton. Assigning a putative function to widely conserved LMBD3 will be of high interest to this completely different audience as well.Reviewer #3 (Evidence, reproducibility and clarity (Required)):In the submitted work "Apical annuli are specialised sites of post-invasion secretion of dense granules in Toxoplasma", the authors explore the role of the apical annuli in *T. gondii*. They identify a number of proteins that localize to the membranes at the annuli, including SNARE proteins that are known players in vesicle fusion. They also shown that knockdown of several annuli localized proteins blocks replication and secretion of dense granule cargo into the parasitophorous vacuole. Overall, the work is well done and an important contribution to the field.Major commentsComment 3.11. In the title and throughout the manuscript the authors claim that the apical annuli are sites of dense granule secretion (e.g. "firmly implicating the apical annuli as the site of dense granule docking and membrane fusion." or "that the apical annuli are sites of vesicle fusion and exocytosis"). However, there does not appear to be direct evidence of the dense granules docking and fusing at these sites.It would be ideal to see vesicles docked via EM at the annuli, either in wildtype or knockdown parasites. This may not be possible – if not, I recommend toning down the conclusions on docking (or "specialized sites of secretion" as this has not been shown) and instead stating that these structures play a critical role in dense granule secretion.

Authors’ response: Please see our response to Comments 2.8 & 2.12, and we have toned down this conclusion as requested to make it clear that direct observations of dense granule fusion are yet to be made. Capturing the transient event of dense granule docking by EM would indeed be a very challenging ambition.

Comment 3.22. The authors should discuss earlier (in the results) the findings of Fu et al. which:

Authors’ response: The parallel study of Fu et al. (2023) has indeed generated some similar data, but there are also multiple points of difference including their conclusions. We discuss all of these relevant points in the Discussion, and believe that it would make the Results narrative confusing to introduce this element of discussion there. Our study has not been performed in response to theirs, but rather was conducted in parallel.

– Show the localization of some of the same SNAREs at the apical annuli. Fu et al. also see localization to the plasma membrane separate from the annuli for some of these proteins. Do you see plasma membrane spots as well upon longer exposures? Can differences be explained by the position or type of tag used?

Authors’ response: Fu et al. have indeed used different reporters and expressed the SNARE fusion proteins with different non-native promoters. They used a very bulky reporter which combined 12 HA tags as well as the large Auxin-Inducible Degron (AID), and together it is possible that they observe some mistargeting artefacts. For our location studies we used the small epitope 3xV5 only. We did not see the additional locations that they report, and this may be due to the larger modification that they made to these proteins.

– Fu et al. also shows similar plaque defects in the knockdowns and loss of trafficking of plasma membrane proteins to the periphery. In general, the studies from this group are very complementary – they should be better acknowledged.

Authors’ response: We have included more frequent reference and comparison to the Fu et al. study now in our Discussion.

– Fu et al. see an invasion defect but no defect in GRA secretion – Do you see an invasion defect? These differences should be discussed

Authors’ response: See our response to Comments 2.1 & 2.13 regarding why the Fu et al. could not detect the GRA secretion defect. We discuss this in our Discussion now (Discussion paragraph four). We also consider the Fu et al. study of an invasion defect as flawed. Both our and their study show that depletion of apical annuli SNAREs has a strong replication phenotype of parasites within the host vacuole. Given induced SNARE depletion must occur during this growing stage of the parasites, to ask if apical annuli could be involved directly in invasion processes requires testing for invasion competence of already very sick cells. It is, therefore, not possible to control for secondary effects on invasion incompetence due to general cell malaise. Furthermore, Fu et al. report on invasion efficiency using an assay that relies on SAG1 presentation on the cell surface. However, they conclude independently in their study that SAG1 delivery to the surface is inhibited in their SNARE knockdowns. This further confounds any attempt to reliable measure a role for these SNAREs in invasion. Therefore, for technical as well as biological reasons, we have not tested for a possible role of annuli in invasion.

– It would be helpful for the field to use the same nomenclature whenever possible. Is it possible to use the naming described earlier?

Authors’ response: Please see our response to Comment 2.4.

Comment 3.33. Figure 1C – The authors use trypsin shaving to demonstrate plasma membrane localization of LMBD3. They are probably correct – but it is important to definitively distinguish between plasma membrane and IMC membrane localization.a. The western blot bands for GAP40 should be quantified. It appears that GAP40 is also reduced and it could be reduced to a similar extent as SAG1 without quantification. In addition, this protection from digestion could be confirmed with a second marker in the space between the PM and IMC membranes like GAP45 (whereas cytoplasmic/mito markers like profilin and Tom40 are likely further protected by the IMC membranes and are thus less relevant here).

Authors’ response: Quantitation of Western blots is notoriously inaccurate and, rather, we use it here as a qualitative indication of trypsin sensitivity of proteins in intact cells. The LMBD3 protein is completely transformed within the first time point (1 hour) to stable products of proteolysis of this polytopic membrane protein — presumably to those now protected within the cell. Known GPI-anchored surface protein SAG1 shows similar immediate sensitivity, although it is known that internalised SAG1 pools are constantly recycled to the surface and hence gradual elimination of the residual SAG1 band over 4 hours. The internal protein markers (GAP40, PRF, TOM40) show no discernible change in the first hour and little if any beyond that (within the variation common to Western blotting). GAP40 shares an equivalent polytopic membrane topology to LMBD3 except it occurs in the IMC membrane directly below the plasma membrane, so we think this is the more suitable control. Thus, this trypsin shaving experiment gives a binary output: sensitive or insensitive. This conclusion is further supported by the published spatial proteomics study (Barylyuk et al. 2020) which shows that LMBD3 segregates with other integral membrane proteins specific to the plasma membrane and not with the IMC proteins. Our super resolution imaging of LMBD3 relative to inner membrane complex markers (Centrin2, GAP45, IMC1) also show it as peripheral to them, further corroborating the plasma membrane location.

b. Is it possible to N-terminally tag LMBD3 and then examine plasma membrane localization by detection of the tag without permeabilization? (this would also confirm the proposed topology)

Authors’ response: We have tried to N-terminally tag LMBD3 with an epitope reporter but this integration was not tolerated by the cell, presumable because it interferes with membrane insertion of this protein that is essential for cell viability. So, this experimental option is not available.

Comment 3.44. I think it is important to make clear for the reader what is happening here. The paper sounds as though the dense granules directly dock at the annuli for release. It also seems possible from this work and Fu et al. that secretion at the annuli occurs via small vesicles that originate from the dense granules. Perhaps a diagram or model would help the reader here (and discuss why DGs or other vesicles are not routinely seen at the annuli if this is the critical portal – and perhaps why the organelles are not clustered in the apical end of the cell if this is where they are needed)

Authors’ response: This comment is related to that of review 2 (Comments 2.8/12), although we note again that Fu et al. did not conclude that dense granules are exocytosed at this site. It is also unclear why this reviewer envisages that small vesicles arise from the dense granules, rather than the dense granule itself fusing at the annuli to the plasma membrane. Indeed, the occurrence of Rab11A on the dense granules, and the accumulation of this protein at the annuli with SNARE knockdown, supports that it is the dense granules that dock at this site. Why dense granules don’t otherwise cluster at their sites of secretion but are instead motile in the cell, their movement driven by Myosin F on actin filaments, is not known. Perhaps these otherwise bulky organelles would create too much cellular crowding that could interfere with other processes. We have addressed all of these points in additions to the discussion so that these interesting unknowns are transparent to the reader (Discussion paragraph 5).

Comment 3.55. Figure 5. The authors state the knockdown results in "strong phenotypes of reduced plaque development".– The plaque assays should be quantified.– Are there no plaques or just very small ones here?

Authors’ response: The reviewer provides no rationale for this request or states what questions could be addressed by doing so. Indeed, none of our conclusions would be affected. We use the plaque assays to test whether each of the proteins tested are independently necessary for some facet of normal parasite growth where the result is binar — no difference in plaque size versus near or complete absence of plaque development. The interpretation of differing plaque sizes between different knockdown mutations is a very inexact science with assumptions of equal rates of protein depletion, sensitivity of relative protein abundance, modes of action of mutation, and kinetics of plaque growth very difficult to validate for meaningful comparisons to be made. Therefore, we don’t see any useful role for plaque quantification in the research questions that we’ve addressed or the conclusions that we present.

Comment 3.66. Figure 6a. Figure 6A – The use of digitonin for semipermeabilization requires controls as there is typically a lot of variability across the monolayer. This is ideally done with something to show that the host plasma membrane has been permeabilized (e.g. host tubulin) and the PVM has not been permeabilized (e.g. SAG1). Otherwise, perhaps the authors could state what percent of cells showed the data like the representative images shown or describe further how selective permeabilization was assessed? (or wider fields with many cells and vacuoles?)

Authors’ response: As requested, we have included a supplemental figure showing wider fields of view where multiple vacuoles are seen. These data show that the vacuoles are similarly stained with no evidence of variability of digitonin permeabilization. The reduction in GRA5 secretion shown by microscopy is further supported by this protein being quantified using proteomics as enriched in the parasites when the apical annuli proteins are depleted (Figure 7).

Comment 3.7b. Figure 6B – "the GRA signal seen within the parasite was increased compared to the control" This is not clear from the AAQb image shown as it appears more is also present in the vacuole (or perhaps residual body?) Can this be clarified?

Authors’ response: Yes, in this image it appears that the ‘residual body’, which is also an integral internal compartment of the growing parasite rosette, is a site of dense granule accumulation. We have modified the text to make it clear that the observations of IFA images showing ‘apparent’ increase in dense granule staining were then directly tested by quantitative proteomics. These subsequent data (Figure 7) provided a clear measure of the increase in dense granule proteins in the parasites when apical annuli function was perturbed.

Minor commentsComment 3.81. Line 215-217 The authors state that "Collectively these data imply that the apical annuli provide coordinated gaps in the IMC barrier that forms at the earliest point of IMC development and that they maintain access of the cytosol to these specialised locations in the plasma membrane."– However, their data shows that LMBD3 only recruits once daughters are emerging (not earliest point of IMC development). Please clarify? Is this just referring to Centrin2 or LMBD3 as well?

Authors’ response: Yes, the other AAPs indicate that these structures form early, and they were mentioned as such in the sentences preceding this statement — hence ‘collectively’.

Comment 3.92. Figure 5. Regarding growth arrest. AAQa appears to show an arrest but is it possible the others just grow slower? Do they arrest later and hence fail to form a plaque? Is there incomplete knockdown which enables a few parasites to persist?

Authors’ response: It is true that it is difficult to discern complete growth arrest from

very retarded growth. However, neither alternative would affect our conclusions where we use these phenotypes as an indication of apical annuli participating in process required for normal growth. All plaque assays show strong growth phenotypes. Nevertheless, we have removed the use of the term ‘growth arrest’ with respect to these phenotypes (including in the Abstract) and replaced it with growth impairment.

Comment 3.103. Line 132, Figure 1 A-C. For clarity it may be better for the reader if LMBD3 is named earlier, or if Figure 1 refers to the gene ID for panels A-C before its named.

Authors’ response: This is a good idea and we have made this change, making note of the rationale for this name when we present the phylogeny.

Comment 3.114. Line 30 – "represent a second structure in the IMC specialised for protein secretion" this is confusing – do the authors mean in addition to the micronemes/rhoptries at the apical complex? Maybe "a second structure in the parasite" would be clearer

Authors’ response: To clarify we have reworded as follows: ‘The apical annuli, therefore, represent a second type of IMC-embedded structure to the apical complex that is specialised for protein secretion’

Comment 3.125. Line 440 – the author states that "these pre- and post-invasion secretion processes are also biochemically separated because both microneme and rhoptry secretion are SNARE-independent" Is this from the Cova and Dubios papers cited a line later? I took a quick scan of these papers and neither appear to show this? Cova claims still this is still unclear and Dubios says SNAREs are likely involved?Authors’ response: While both microneme and rhoptry secretion use distinctive molecular machineries for controlling membrane fusion for exocytosis, it is true that it is not formally known that these processes completely lack SNARE involvement, and neither paper cited here can eliminate this possibility. We have therefore, removed this short part of the discussion where we consider that dense granules might be unique amongst these three compartments in relying on SNAREs.Text editingComment 3.131. Line 94 – plasma membrane or cell surface. Clarify here – do you mean plasma membrane or under the membrane at the periphery?

Authors’ response: We have modified as: ‘plasma membrane including the cell surface’.

Comment 3.142. Line 321 refers to Figure 6A but should say 7A. Panel 7B is never referenced in the text.

Authors’ response: Thank you, we have corrected this and only sited Fig7 because A and B are both relevant to the statement made in the text.

Comment 3.153. Line 347-242 and Figure 4A – the discussion of Q-SNARES and diagram could use some references for the reader

Authors’ response: Thank you for this suggestion, we have acted on this request.

Comment 3.164. The methods says plaque assays were 7 days, Figure 5 legend says 8 days

Authors’ response: Thank you, this is corrected as 8 days.

Referees cross-commenting– I completely agree with Rev 2– I also think examining invasion given Rev1 comment on the micronemes and the data from Fu et al. would be worthwhile and straightforward to do

Authors’ response: Please see our response to Comment 3.2 where the validity of measuring invasion competence of poorly growing, and/or arrested, parasites is scientifically questionable. It would require controls of similarly unhealthy parasites where the apical annuli are unaffected, but it is difficult to imagine how one would deliver such a control.

Reviewer #3 (Significance (Required)):This is an excellent study that assesses the role of apical annuli in parasite secretion. It is an important addition to the field (and outstanding imaging that provides a high level of detail to the study). The study could be improved by better integrating a recent similar study noted by the authors and in the review

Authors’ response: We have provided more direct discussion of the Fu et al. paper in our Discussion section.